# CPSample: Classifier Protected Sampling for Guarding Training Data During Diffusion

**Joshua Kazdan[1], Hao Sun[2], Jiaqi Han[2], Felix Petersen[2], Frederick Vu[3], Stefano Ermon[2]**

[1]Department of Statistics, Stanford University
[2]Department of Computer Science, Stanford University
[3]Department of Mathematics, UCLA

## Abstract

Diffusion models have a tendency to exactly replicate their training data, especially when trained on small datasets. Most prior work has sought to mitigate this problem by imposing differential privacy constraints or masking parts of the training data, resulting in a notable substantial decrease in image quality. We present CPSample, a method that modifies the sampling process to prevent training data replication while preserving image quality. CPSample utilizes a classifier that is trained to overfit on random binary labels attached to the training data. CPSample then uses classifier guidance to steer the generation process away from the set of points that can be classified with high certainty, a set that includes the training data. CPSample achieves FID scores of 4.97 and 2.97 on CIFAR-10 and CelebA-64, respectively, without producing exact replicates of the training data. Unlike prior methods intended to guard the training images, CPSample only requires training a classifier rather than retraining a diffusion model, which is computationally cheaper. Moreover, our technique provides diffusion models with greater robustness against membership inference attacks, wherein an adversary attempts to discern which images were in the model's training dataset. We show that CPSample behaves like a built-in rejection sampler, and we demonstrate its capabilities to prevent mode collapse in Stable Diffusion.

## 1 Introduction

Diffusion models are an emerging method of image generation that have surpassed GANs on many common benchmarks (Dhariwal & Nichol, 2021), achieving state-of-the-art FID scores on CIFAR-10 (Krizhevsky, 2009), CelebA (Liu et al., 2015), ImageNet (Deng et al., 2009), and other touchstone datasets. Although their capabilities are impressive, diffusion models still suffer from the tendency to exactly replicate images found in their training sets (Carlini et al., 2021; Jagielski et al., 2023; Somepalli et al., 2023a). This problem is especially pronounced when the training set contains duplicates (Webster et al., 2023). Given that diffusion models are sometimes trained on sensitive content, such as patient data (Kazerouni et al., 2023; Pinaya et al., 2022) or copyrighted data (Dhariwal & Nichol, 2021), this behavior is generally unacceptable. Indeed, Google, Midjourney, and Stability AI are already facing lawsuits for using copyrighted data to train image generation models (Brittain, 2023; 2024), some of which exactly replicate images from their training data during inference (Marcus & Southen, 2024).

The strongest formal guarantee against replicating or revealing training data is differential privacy (DP) (Dwork & Roth, 2014). Unfortunately, differential privacy is at odds with generation quality. Although differentially private training has been implemented for GANs (DP-GAN) (Xie et al., 2018), diffusion models (DPDM, DP-Diffusion) (Dockhorn et al., 2023; Ghalebikesabi et al., 2023), and latent diffusion models (DP-LDM) (Lyu et al., 2024), it typically results in significant degradation of image quality. Moreover, one cannot easily make a pretrained model differentially private, implying that to achieve differential privacy, one must retrain from scratch. This makes negotiating the trade-off between privacy and quality challenging, as trying different levels of privacy requires retraining. Due to the difficulty of achieving differential privacy while simultaneously maintaining quality, some researchers have pursued more attainable model characteristics that have

the same flavor as differential privacy. A frequent benchmark for privacy is robustness to membership inference attacks (Hu & Pang, 2023), whereby the attacker aims to infer whether a given image was used to train the model. Although researchers have devised a multitude of loss and likelihood-based membership inference attacks, so far, there are few existing methods that explicitly aim to defend against these attacks besides differential privacy and data augmentation (Matsumoto et al., 2023b; Pang et al., 2023). A second privacy benchmark measures the cosine similarity in a feature space of a generated image to its nearest neighbor in the training data (Daras et al., 2024; Douze et al., 2024). Ambient diffusion (Daras et al., 2024) is one method to prevent excessive similarity to the training data without enforcing differential privacy; however, ambient diffusion still has notable negative effects on FID scores.

Until recently, preventing image replication by diffusion models has involved various forms of data corruption during training, either by adding noise to gradients (Abadi et al., 2016), diversifying images and captions (Somepalli et al., 2023b), or corrupting the images themselves (Daras et al., 2024). Hyperparameter tuning for these methods requires retraining, making it difficult to calibrate them to the necessary level of privacy. Simple alternatives, like rejection sampling, are effective because they can guarantee that the training images will not be exactly replicated. However, standard rejection sampling has major drawbacks, too. For instance, rejection sampling redistributes probability mass in an inefficient way and requires resampling, which can decrease speed. In extreme cases of mode collapse such as those uncovered by Webster (2023), Stable Diffusion must be queried dozens or even hundreds of times before producing an original image, making rejection sampling impractical. Rejection sampling is also prone to membership inference attacks and privacy leakages (Awan & Rao, 2023).

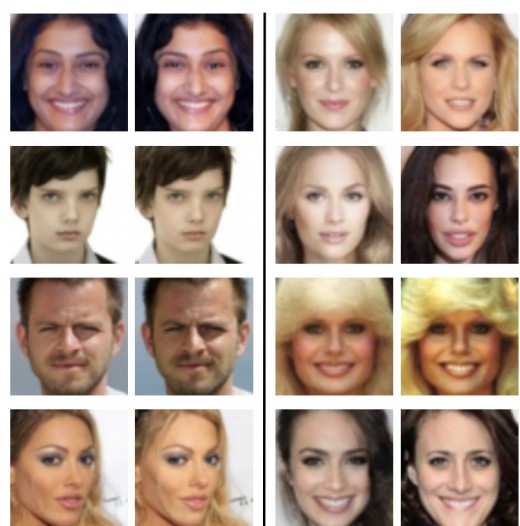

Figure 1: Generated image and most similar training image pairs for DDIM sampling (left) and CPSample with $\alpha=0.001$, $s=1\,000$ (right). We sample 100 images and display the four with the highest similarity to their nearest neighbors in the training data.

We present classifier-protected sampling (CPSample), a diffusion-specific data protection technique that, while not strictly differentially private, fortifies against some membership inference attacks and greatly reduces excessive similarity between the training and generated data. CPSample is computationally more efficient than existing training-based methods of improving privacy for diffusion models. The basic idea is to overfit a classifier on random binary labels assigned to the training data and use this classifier during sampling to guide the images away from the training data. We show that our method has an effect similar to rejection sampling while removing or reducing the need to resample. Unlike rejection sampling, CPSample offers protection against some membership inference attacks during the generation process rather than only protecting the end product. CPSample achieves SOTA image quality, improving over previous data protection methods, such as ambient diffusion, DPDMs, and PAC Privacy Preserving Diffusion Models (Xu et al., 2023) for similar levels of "privacy." Unlike most other methods designed to shield the training data, one can simply adjust the level of protection provided by CPSample without retraining the classifier used for guidance. CPSample is applicable to existing image models without any expensive retraining of the diffusion models. We summarize the primary contributions of our work as follows:

- In Section 3.1, we introduce CPSample, a novel method of classifier-guidance for privacy protection in diffusion models that can be applied to existing models without retraining.

- We show theoretically in Section 3.2 and empirically in Section 4.1 that CPSample prevents training data replication in unguided diffusion. We also provide evidence in Section 4.2 that CPSample can protect text-based image generation models, like Stable Diffusion.

- We give empirical evidence that CPSample can foil some membership inference attacks in Section 4.3.
- We demonstrate in Section 4.4 that CPSample attains better FID scores than existing methods of privacy protection while still eliminating replication of the training data.

## 2 BACKGROUND AND RELATED WORK

### 2.1 DIFFUSION MODELS

We begin with a review of diffusion models. Denoising diffusion probabilistic models (DDPMs) (Sohl-Dickstein et al., 2015; Ho et al., 2020) gradually add Gaussian noise to image data during the "forward" process. Meanwhile, one trains a "denoiser" to predict the original image from the corrupted samples in a so-called "backward" process. During the forward process, one assigns

$$x_t = \sqrt{\alpha_t}x_0 + \sqrt{1 - \alpha_t}\epsilon \tag{2.1}$$

where $\epsilon \sim \mathcal{N}(0, I)$, $x_0$ is the original image, and $\alpha_t$ indicates the noise schedule. The variable $t \in \{0, ..., T\}$ specifies the step of the forward process, where $x_0$ represents an image in the training data. When $\alpha_T$ is set sufficiently close to 0, $x_T$ is approximately drawn from a standard normal distribution. During intermediate steps, the distribution of $x_t$ is

$$q(x_t \mid x_0) = \mathcal{N}(x_t; \sqrt{\alpha_t}x_0, (1 - \alpha_t)I). \tag{2.2}$$

During training, one performs gradient descent on $\theta$ to minimize the score-matching loss, given by

$$\mathbb{E}_{\epsilon \sim \mathcal{N}(0,1), x_0 \sim \mathcal{D}}\left[\sum_{t=1}^{T} \frac{1}{2\sigma_t^2} \|\epsilon - \epsilon_\theta(\sqrt{\overline{\alpha}_t}x_0 + \sqrt{1 - \overline{\alpha}_t}\epsilon, t)\|^2\right]. \tag{2.3}$$

Here, $\mathcal{D}$ is the target distribution, which is approximated by sampling from the training data. Finally, to generate a new image, one samples standard Gaussian noise $x_T \sim \mathcal{N}(0, I)$. Then, one gradually denoises $x_T$ by letting

$$x_{t-1} = \frac{1}{\sqrt{\alpha_t}}\left(x_t - \frac{1 - \alpha_t}{\sqrt{1 - \overline{\alpha}_t}}\epsilon_\theta(x_t, t)\right) + \sigma_t z_t, \tag{2.4}$$

where in each step, one has $z_t \sim \mathcal{N}(0, I)$, and $\sigma_t$ and $\alpha_t$ are scalar functions determined by the noise schedule that govern the rate of the backward diffusion process.

Despite the superior image quality afforded by DDPMs, the sampling process sometimes involves 1 000 or more steps, which has led to a variety of sampling schemes and distillation methods for speeding up inference (Song et al., 2021; Kim et al., 2024; Song et al., 2023; Gu et al., 2023). One of the most commonly used modifications to the sampling process is denoising diffusion implicit models (DDIMs), which enable skipping steps in the backward process.

Currently, the state-of-the-art for guided generation is achieved by models with classifier-free guidance (Ho & Salimans, 2022). However, since CPSample employs a classifier to prevent replication of its training data, it is more useful for us to review its predecessor, classifier-guided diffusion (Lim et al., 2023; Dhariwal & Nichol, 2021). In classifier guided diffusion, a pretrained classifier $p_\phi(y \mid x_t, t)$ assigns a probability to the event that $x_t = \sqrt{\alpha_t}x_0 + \sqrt{1 - \alpha_t}\epsilon$ for some $x_0$ with label $y$. The sampling process for classifier-guided DDIM is modified by

$$\hat{\epsilon}_t = \epsilon_\theta(x_t) - \sqrt{1 - \overline{\alpha}_t}\nabla_{x_t} \log p_\phi(y \mid x_t, t) \tag{2.5}$$

$$x_{t-1} = \sqrt{\overline{\alpha}_{t-1}}\left(\frac{x_t - \sqrt{1 - \overline{\alpha}_t}\hat{\epsilon}_t}{\sqrt{\overline{\alpha}_t}}\right) + \sqrt{1 - \overline{\alpha}_t}\hat{\epsilon}_t. \tag{2.6}$$

Such a modification of the sampling procedure corresponds to sampling $x_t$ from the joint distribution:

$$p_{\theta,\phi}(x_t, y \mid x_{t+1}, t) = Zp_\theta(x_t \mid x_{t+1}, t)p_\phi(y \mid x_t, t) \tag{2.7}$$

where $Z$ is a normalization constant. This formulation can be adapted for continuous-time models, but for discrete-time models, additional care must be taken to ensure accuracy (see Appendix A for additional details).

## 2.2 Privacy in Diffusion Models

Differential privacy (DP) is generally considered to be the gold standard for protecting sensitive data. The formal definition of $(\varepsilon\text{-}\delta)$ differential privacy is as follows (Dwork et al., 2006; Dwork & Roth, 2014):

**Definition 2.1** $((\varepsilon\text{-}\delta)$-Differential privacy). *Let $\mathcal{A}$ be a randomized algorithm that takes a dataset as input and has its output in $\mathcal{X}$. If $D_1$ and $D_2$ are datasets with symmetric difference $1$, then $\mathcal{A}$ is $\varepsilon\text{-}\delta$ differentially private if for all $S \subset \mathcal{X}$,*

$$\mathbb{P}(\mathcal{A}(D_1) \in S) \leq \mathbb{P}(\mathcal{A}(D_2) \in S)e^{\varepsilon} + \delta. \tag{2.8}$$

DP ensures that the removal or addition of a single data point to the dataset does not significantly affect the outcome of the algorithm, thus protecting the identity of individuals within the dataset. Existing DP diffusion models (Dockhorn et al., 2023; Ghalebikesabi et al., 2023; Lyu et al., 2024) achieve DP through DP stochastic gradient descent (DP-SGD) (Abadi et al., 2016), in which Gaussian noise is added to the gradients during training.

Though DP offers a formal guarantee that one's data is secure, imposing a DP constraint in practice severely compromises the quality of the synthetic images. Therefore, researchers have largely resorted to demonstrating that models exhibit various relaxations of strict DP (Vyas et al., 2023), such as Probably Approximately Correct DP (PAC-DP) (Xiao & Devadas, 2023) or other empirical metrics of privacy.

One common approach is to measure the similarity between a generated image and its nearest neighbors in the training dataset with the aim of minimizing the number of highly similar neighbors (Daras et al., 2022; 2024). Typically, similarity is quantified using either Euclidean distance or cosine similarity, which is given by

$$\frac{x^T \cdot \mathrm{C}(x)}{\|x\| \cdot \|\mathrm{C}(x)\|}, \tag{2.9}$$

with $\mathrm{C}(x)$ denoting the nearest neighbor of $x$ among the training data, and which is typically computed in a feature space rather than the raw pixel space (Nguyen & Bai, 2011; Xia et al., 2015). Tools have been developed to help compute nearest neighbors efficiently, since naïve pairwise comparisons are too computationally expensive. In 2023, MetaAI developed the FAISS library for efficient similarity search using neural networks (Douze et al., 2024), making this type of privacy metric possible to compute approximately in a reasonable amount of time. Empirically, for CIFAR-10, we observed that images with similarity scores above $0.97$ were nearly identical, whereas for CelebA, the threshold was approximately $0.95$, and for LSUN Church, images with similarity above $0.90$ were sometimes, though not always, nearly identical.

Ambient diffusion reduces similarity to the nearest neighbor by masking pixels during training and only scoring the model based on the visible pixels. In this way, the model never has access to a full, uncorrupted image and is therefore less likely to replicate full images (Daras et al., 2024). The downside is that masking pixels, even at the relatively modest rate of $20\%$, leads to notable image quality degradation as measured via the FID score. Moreover, ambient diffusion shifts the entire distribution of similarity scores towards lower similarity, whereas we should ideally prevent the generation of images with high similarity to the training data while leaving the rest of the distribution untouched.

Until recently, all attempts at enforcing privacy for diffusion models occurred during training. In 2023, Xu et al. (2023) developed a method of classifier-guided sampling (PACPD) that has PAC privacy advantages over standard denoising. For text-guided models, Somepalli et al. (2023b) developed a method of randomly changing influential input tokens to avoid exact memorization, and Wen et al. (2024) protected training data using a regularization technique on the classifier-free guidance network during training. Golatkar et al. (2024) used a mixture of public and private images to prevent copyright infringement, and Golatkar et al. (2023) developed "compositional diffusion models" to customize data access across different groups. Recently,Chen et al. (2024) devised a guidance method (AMG) which calculates similarity metrics at each step in the denoising schedule in order to guide the sampling process away from similar data points in the training corpus. By utilizing similarity metrics directly, they were able to effectively eliminate memorization in both text-conditional and unconditional diffusion models. Though theoretically valuable, the need to have access to the training data– or at least to embeddings of the training data points– and to compute similarity measures at runtime is impractical for use outside of a research environment.

## 2.3 MEMBERSHIP INFERENCE ATTACKS

A third privacy measurement comes from membership inference attacks (Dubiński et al., 2024; Pang & Wang, 2023; Duan et al., 2023; Wu et al., 2022), whereby one tries to discern whether a given data point was a member of the training set for the model. Robustness to membership inference attacks is implied by differential privacy. Membership inference attacks against diffusion models usually hinge on observed differences in reconstruction loss or likelihood that come from overfitting. In this paper, we will use a slight modification of the membership inference attack from Matsumoto et al. (2023a) as described in Algorithm 1 in Appendix E.

## 3 PROTECTING PRIVACY DURING SAMPLING

In this section, we address the problem of training data replication in diffusion models, which poses significant privacy risks. One common solution to this problem is rejection sampling, whereby samples that closely resemble training data are discarded. However, rejection sampling has several shortcomings: it is computationally expensive and inefficient, and it only provides protection in the final output, not during the sampling process itself. Moreover, in extreme cases of mode collapse, one may need to generate dozens of images before generating original content when using rejection sampling.

To overcome these limitations, we introduce CPSample, a method that reproduces some of the benefits of rejection sampling without the need for resampling. CPSample integrates classifier guidance into the sampling process to steer the generation away from the training data. We overfit a classifier on random binary labels assigned to the training data and use this classifier during sampling to adjust the generated images, thereby reducing the likelihood of replicating training data while preserving image quality.

### 3.1 SAMPLING METHOD

The first step in CPSample is to train a network that can provide information about how likely a sample $x_t$ is to turn into a member of the training data at the end of the denoising process. For this task, we use a classifier trained to memorize random binary labels assigned to the training data. It was shown in Zhang et al. (2017) that this can be achieved with a network with a number of parameters that is linearly proportional to the size of the dataset, with a small constant of proportionality. Additionally, the training time required to memorize random labels is only a small constant factor more compared to the time it takes to memorize real, non-random labels.

To address duplicated data in the training corpus, after the classifier has been sufficiently trained, items for which the classifier still shows significant loss can be reassigned a common label. Further training then ensures the classifier memorizes these items.

During the denoising process, whenever the classifier predicts a label $y \in \{0, 1\}$ for $x_t$ with probability greater than $1 - \alpha$, we perturb $x_{t-1}$ towards the opposite label using classifier guidance. For example, if the classifier predicts the label 1 with high probability, we employ classifier guidance to adjust the sampling process to draw from the conditional distribution $p_{\theta,\phi}(x_{t-1} \mid x_t, t, y = 0)$, reducing the likelihood of the generated sample being close to the training data.

To state our procedure more precisely, let $\epsilon_\theta(\cdot, \cdot)$ be the denoiser. Note that the classifier is trained only once on the training data and not during each sample generation. The sampling process is then modified in the following steps:

1. Randomly assign Bernoulli(0.5) labels to each member of the training data, and let $B \in \{0, 1\}^n$ index these random labels. Train a classifier $p_\phi(y \mid x_t, t)$ to predict these labels. Here, $x_t$ is generated by corrupting the training data $x_0$ with noise: $x_t = \sqrt{\alpha_t}x_0 + \sqrt{1 - \alpha_t}\epsilon$ for $\epsilon \sim \mathcal{N}(0, I)$ and $t \in \{0, ..., T\}$.

2. Set a tolerance threshold $0 < \alpha < 0.5$ and a scale parameter $s$. Let $p_\phi(y \mid x_t, t)$ be the probability assigned to the label $y$ by the classifier $p_\phi(y \mid x_t, t)$. Sample $x_T \sim \mathcal{N}(0, I)$. For $t \in \{T, ...., 1\}$, if $p_\phi(y = 0 \mid x_t, t) < \alpha$, replace $\epsilon_\theta(x_t, t)$ with

$$\hat{\epsilon}_{\theta,\phi}(x_t, t) = \epsilon_\theta(x_t, t) - s\sqrt{1 - \overline{\alpha}_t} \cdot \nabla \log(p_\phi(y = 0 \mid x_t, t)).$$

If $p_\phi(y = 1 \mid x_t, t) < \alpha$, replace $\epsilon_\theta$ with

$$\hat{\epsilon}_{\theta,\phi}(x_t, t) = \epsilon_\theta(x_t, t) - s\sqrt{1 - \overline{\alpha}_t} \cdot \nabla \log(p_\phi(y = 1 \mid x_t, t)).$$

Otherwise, we leave the sampling process unchanged.

Though the choice of random labels for the classifier initially may seem counter-intuitive, it has several advantages over other approaches. If we used labels corresponding to real attributes of the data, the classifier would influence the content of the generated images in ways that could compromise their authenticity and diversity. This is because the guidance would push the generated images towards or away from specific attributes, altering the intended distribution. The perturbation applied by the gradient of the log probability in CPSample moves the generated images away from regions where they can be easily classified as similar to the training data. This method is more effective than adding random noise, which would require a significant amount of noise to achieve the same effect, thus degrading image quality.

Unlike past training-based methods of privacy protection such as ambient diffusion and DPDM, once we have trained the classifier, we can adjust the level of protection by tuning the hyperparameters $s$ and $\alpha$ without necessitating retraining of the classifier or denoiser. Our method also does not require access to the training data or excessive additional computation during sampling as the inferenced-based method AMG does.

## 3.2 THEORY

In this section, we show that CPSample functions similarly to rejection sampling when preventing exact replication of the training images. We work under the following assumptions:

**Assumption 1.** *Suppose that the classifier $p_\phi(y \mid x, t)$ has Lipschitz constant $L$ in the argument $x$ with respect to a metric $d(\cdot, \cdot) : \chi \times \chi \to \mathbb{R}_{\geq 0}$, where $\chi$ denotes the image space.*

**Assumption 2.** *Let $y_i$ be the random label assigned to $x_i \in D$, where $D$ is the training data. Let $\kappa < \frac{1}{2}$ be such that for all $x_i \in D$, we have*

$$p_\phi(y_i \mid x_i, 0) \in (1 - \kappa, 1]. \tag{3.1}$$

**Assumption 3.** *Suppose that CPSample generates data $\tilde{x}$ such that $\lambda < p_\phi(y \mid \tilde{x}, 0) < 1 - \lambda$ with probability greater than $1 - \nu$, where we are able to govern $\nu$ and $\lambda$ by adjusting $s$ and $\alpha$ in Section 3.1.*

In Assumption 1 the constant $L$ can be difficult to evaluate, but the assumption holds for neural network classifiers. Methods exist that can bound the local Lipschitz constant around the training data (Huang et al., 2021), which one can use to strengthen the guarantees of Lemma 1. Assumption 3 holds well empirically, and in Assumption 2, one can typically exert strong control over the size of $\kappa$ without incurring too much additional computational overhead Zhang et al. (2017). Concretely, we were able to train our classifier to have a cross-entropy loss below $0.05$ in the experiments from Sections 4.1 and 4.2. Moreover, during sampling, we observed that CPSample had control over the quantity $p_\phi(y \mid x_t, t)$. An example is given in Figure 7.

Given these assumptions, we can demonstrate the following simple lemma, which links the behavior of CPSample to that of a rejection sampler without requiring expensive comparisons to the training dataset. A proof can be found in Appendix A. We note that the assumptions are admittedly relatively strong, as Lemma 1 is designed primarily to give theoretical intuition for how CPSample works rather than practical guarantees on how well it protects the data. We refer to reader to our empirical results (See Section 4) for practical demonstrations of the method's efficacy.

**Lemma 1.** *Under the above assumptions, choose $\varepsilon > 0$ and $0 < \delta < \frac{\frac{1}{2} - \kappa}{L}$. Setting $\nu = \varepsilon$ and $\lambda = \kappa + L\delta$, when drawing a single sample, with probability greater than $1 - \varepsilon$, CPSample generates an image that lies outside of $S = \bigcup_{x \in D} B_\delta(x)$ in the metric space defined by $d$.*

Note that the ability to control $\mathbb{P}\left(\tilde{x} \in \bigcup_{x \in D} B_\delta(x)\right)$ gives the same guarantee offered by rejection sampling. However, in extreme instances of mode collapse such as those exhibited by Stable Diffusion in Section 4.2, one might have to resample hundreds of times to generate original images, making standard rejection sampling highly inefficient. CPSample is able to produce original images without this high level of inefficiency.

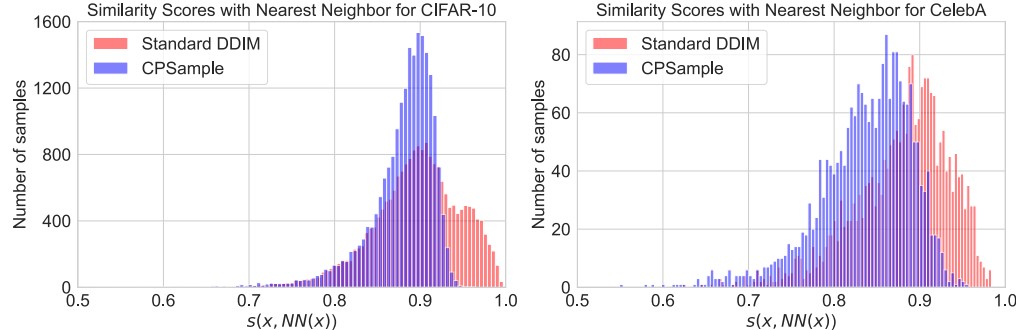

Figure 2: Cosine similarity in feature space between generated images and their nearest neighbor in the fine-tuning dataset for standard DDIM sampling (red) and CPSample (blue) with $\alpha = 0.001, s = 1$ on CIFAR-10 (left) and with $\alpha = 0.001, s = 1\,000$ on CelebA-64 (right). Similarity scores were computed for $21\,000$ generated samples for CIFAR-10 and $8\,000$ images for CelebA. Note that standard DDIM exhibits many more samples with similarity scores exceeding the thresholds from Table 1.

## 4 EMPIRICAL RESULTS

We run three distinct sets of experiments to demonstrate the ways in which CPSample protects the privacy of the training data. First, we statistically test the ability of CPSample to reduce similarity between generated data and the training set for unguided diffusion. We then demonstrate that CPSample can prevent Stable Diffusion from generating memorized images. Finally, we measure robustness against membership inference attacks. Hyperparameters, in all empirical tests, are chosen to maximize image quality while eliminating exact matches. In our tests of image quality, we find that CPSample far outperforms existing methods of protecting the training data.

### 4.1 SIMILARITY REDUCTION

We generate images using DDIM with CPSample and $1\,000$ denoising steps. The nearest neighbor to each generated image is found using Meta's FAISS model (Douze et al., 2024). Similarity between two images is measured by cosine similarity in a feature space defined by FAISS. We empirically find that a similarity score exceeding 0.97 often indicates nearly-identical images for CIFAR-10. For CelebA and LSUN Church, the thresholds lie around 0.95 (Daras et al., 2024) and 0.90, respectively. Note that a cosine similarity score above the thresholds given is a necessary but not sufficient condition for images to look very alike. To ensure that we can observe a larger number of images with similarities exceeding our thresholds, we fine-tune the models using DDIM (Song et al., 2021) on a subset of the data that consisted of $1\,000$ images, as was done in Daras et al. (2024). This modification allows us to statistically test the efficacy of CPSample without the large number of samples required to do hypothesis testing on rare exact replication events. After fine-tuning, up to $12.5\%$ of the images produced by unprotected DDIM are nearly exact replicates of the fine-tuning data. One can see from Table 1 that CPSample significantly reduces the fraction of generated images that have high cosine similarity to members of the fine-tuning set. One can see histograms of the similarity score distribution with and without CPSample in Figures 2 and 10. Figures 1 and 3 show the most similar pairs of samples and fine-tuning data points. Uncurated images generated from CPSample can be found in Appendix F. While CPSample effectively reduces the similarity between generated images and the training data, our results in Table 4 indicate that CPSample achieves minimal degradation in quality compared to previous methods.

### 4.2 STABLE DIFFUSION

As a second demonstration of CPSample, we present evidence that CPSample can prevent well-known examples of mode collapse in near-verbatim attacks against Stable Diffusion (Webster, 2023; Wen et al., 2024). We curate a small dataset of commonly reproduced images (Somepalli et al., 2023b) and include other images from the LAION dataset depicting the same subjects, while ensuring that this dataset contains no duplicates. In this more targeted application, CPSample can prevent exact replication when used with the right hyperparameters. See Figure 6 and Table 2 for more details.

| DDIM (Unprotected) | CPSample (Protected) | DDIM (Unprotected) | CPSample (Protected) |
|---|---|---|---|

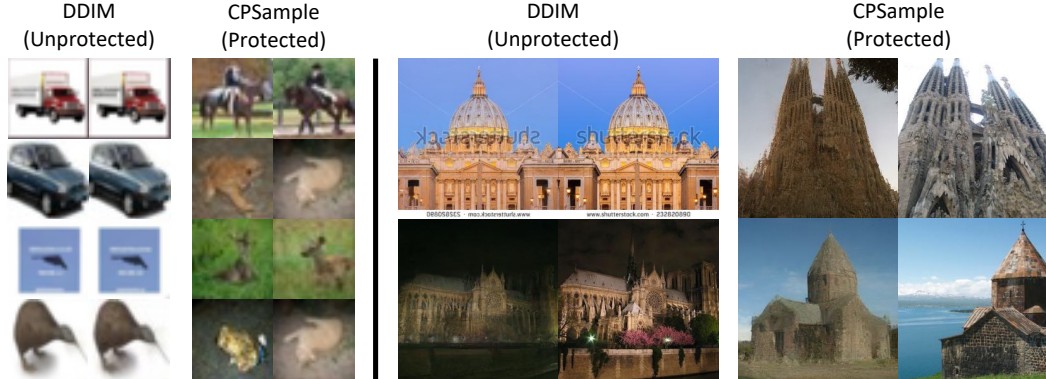

Figure 3: Generated images and their most similar training image pairs for DDIM sampling and CPSample with $\alpha = 0.001$, $s = 1$ on CIFAR-10 (left) and $\alpha = 0.1$, $s = 10$ on LSUN Church (right). For each pair, the image on the left is the generated sample, and the one on the right is its nearest neighbor in the training set. These are the four examples out of 21 000 images on CIFAR-10 and two out of 1 700 images on LSUN Church with the highest similarity scores with their nearest neighbor.

Table 1: Reduction in cosine similarity between generated images and nearest neighbor in fine-tuning data. [1] $p$-values were computed using a $\chi^2$ test for the null hypothesis $H_0$: "CPSample did not reduce the fraction of images with similarity score exceeding the threshold."

| Dataset | FT Steps | $\alpha$ | Scale | Threshold | DDIM | CPSample | p-value[1] |
|---|---|---|---|---|---|---|---|
| CIFAR-10 | 150k | 0.001 | 1 | 0.97 | 6.25% | 0.00 % | <0.0001 |
| CelebA | 650k | 0.001 | 1 000 | 0.95 | 12.5% | 0.10% | <0.0001 |
| LSUN Church | 455k | 0.1 | 10 | 0.90 | 0.73% | 0.04% | 0.013 |

Although CPSample does not provide as robust protection in this setting compared to Somepalli et al. (2023b); Wen et al. (2024), these results highlight its potential for data protection in text-guided diffusion models. Moreover, the methods developed in Somepalli et al. (2023b); Wen et al. (2024) do not apply to unguided diffusion models. This application also highlights how CPSample can protect only a private subset of the training data, for which it does not require access to the entire training dataset.

Table 2: Details of generation on Stable Diffusion.

| Image | Original caption | Modified caption | $\alpha$ | scale | guidance |
|---|---|---|---|---|---|
| A | "Rambo 5 and Rocky Spin-Off - Sylvester Stallone gibt Updates" | "Rocky and Rambo Spin-Off - Sylvester Stallone gibt Updates" | 0.5 | 2 000 | 1.5 |
| B | "Classic cars for sale" | "Classic car for sale" | 0.3 | 100 | 1.5 |
| C | "Red Exotic Fractal Pattern Abstract Art On Canvas-7 Panels" | "Red Exotic Fractal Pattern Abstract Art On Canvas-7 Panels" | 0.5 | 2 000 | 1.5 |

### 4.3 Membership Inference Attacks

We also assess CPSample's ability to protect against membership inference attacks. Following Algorithm 1, we compute the mean reconstruction error for the training and test datasets and determine whether there is a statistically significant difference. To evaluate resistance to inference attacks, we use a model trained on the entire set of 50 000 CIFAR-10 training images. We compare the reconstruction loss on these 50 000 training images to the reconstruction loss on the 10 000 withheld test samples included in the CIFAR-10 dataset. We compare the difference in reconstruction loss between these two datasets both for CPSample, using a classifier trained on the entirety of the CIFAR-10 training data with random labels, and for standard DDIM sampling. We demonstrate CPSample's resistance to inference attacks for $\alpha \in \{0.5, 0.25, 0.001\}$ over approximately 8000

images from each of the training and test datasets. The $p$-values in this experiment are based on a two-sample, single-tailed $Z$-score that tests the null hypothesis "the average training reconstruction loss is less than or equal to the average test reconstruction loss." Precisely, let $n$ denote the number of training data points and $m$ denote the number of sampled test data points. The test statistic is then given by

$$\frac{\mu_{\text{test}} - \mu_{\text{train}}}{\sqrt{V_{\text{test}}/m + V_{\text{train}}/n}}.$$

Here, the symbol $V$ indicates the variance and $\mu$ indicates the mean. In this context, failure to reject the null hypothesis indicates a success for CPSample.

We observe that in our experiments, a very low value of $\alpha$ leads to a higher $p$-value, which is counterintuitive on first glance. However, we believe that this occurs due to the fact that a small value of $\alpha$ results in a more targeted application of CPSample, driving the loss up exclusively around the training data points. As shown in Table 3 for values of $\alpha$ between 0 and 0.5, a conclusive membership inference attack against CPSample is not possible. We provide a second black-box membership inference attack based on permutation testing in Appendix E.

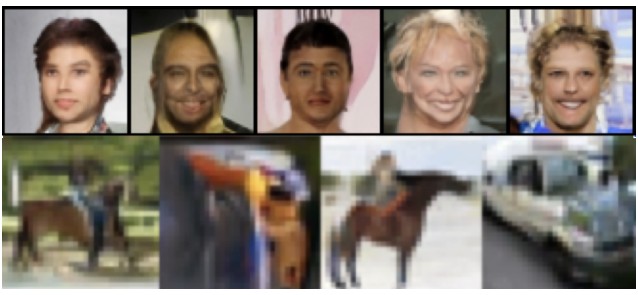
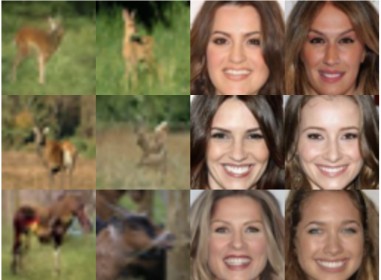

Figure 4: Uncurated samples from DP-LDM (Lyu et al., 2024) on CelebA (Top) and Ambient Diffusion (Daras et al., 2024) with corruption 0.2 (Bottom). These samples are of noticably lower quality than the uncurated samples from CPSample, found in Appendix F.

Figure 5: The generated and real images with the highest similarity for CIFAR-10 (left) and CelebA (right) out of $50\,000$ samples used to compute FID score.

## 4.4 QUALITY COMPARISON

As mentioned in the introduction and Section 4.1, other methods of privacy protection suffer from severe degradation of quality as measured by FID score. Here, we provide an FID score comparison between the CPSample model fine-tuned on curated subsets of CIFAR-10 and CelebA and existing methods of privacy protection. FID scores for unconditional generation of CIFAR-10 and CelebA are presented in Table 4. The images with the highest similarity to the training set, determined using FAISS, are shown in Figure 5. The particular values of $\alpha$ and $s$ were set in an attempt to find the least aggressive settings that still completely prevent exact replication of the training data. FID scores over a grid search on the hyperparameters $\alpha$ and $s$ are displayed in Table 7. We include images from DP-LDM (Lyu et al., 2024) and Ambient Diffusion (Daras et al., 2024) in Figure 4 to emphasize the gains in quality.

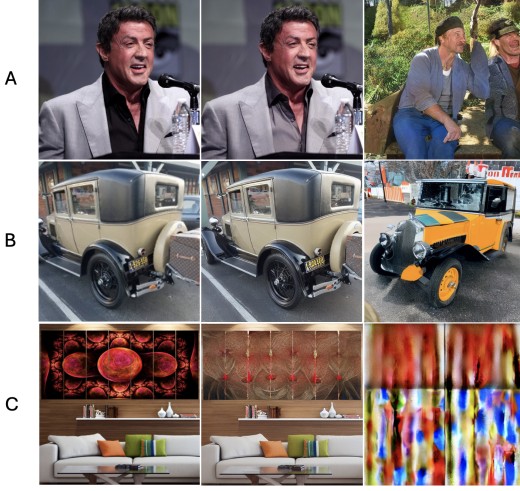

Figure 6: Selected examples for Stable Diffusion: original image (left), image generated from a similar caption by Stable Diffusion v1.4 (center), image generated with CPSample (right).

Table 3: Difference in mean reconstruction error between train and test data for CIFAR-10.

| Method | Test statistic | p-value |
|---|---|---|
| DDIM | 138 | $\approx 0$ |
| Ambient (Corruption 0.2) | 0.141 | 0.44 |
| Ambient (Corruption 0.8) | -0.024 | 0.51 |
| CPSample ($\alpha = 0.5$) | 0.59 | 0.28 |
| CPSample ($\alpha = 0.25$) | 0.23 | 0.41 |
| CPSample ($\alpha = 0.001$) | -0.86 | 0.81 |

Table 4: FID score comparison on the CIFAR-10 and CelebA datasets.

| Method | CIFAR-10 | CelebA |
|---|---|---|
| DDIM | 3.17 | 1.27 |
| Ambient (Corruption 0.2) | 11.70 | 25.95 |
| DPDM ($\epsilon = 10$) | 97.7 | 78.3 |
| DP-Diffusion ($\epsilon = 10$) | 9.8 | - |
| DP-LDM ($\epsilon = 10$) | 8.4 | 16.2 |
| CPSample ($\alpha = 0.001, 0.05$) | **4.97** | **2.97** |

## 5 LIMITATIONS

As mentioned in Section 3.1, the difference in training time required to get a classifier to memorize random labels versus real labels has been shown to be only a small constant factor Zhang et al. (2017). Compared to other leading methods of protecting training data, such as ambient diffusion, DPDM, and AMG, our method is significantly easier to employ in terms of computational resources. However, as we lack the resources to provide further empirical evidence beyond what has already been demonstrated in the literature, we leave this remark as a flag for a potential practical limitation of our method.

Of slight theoretical concern is the difficulty in providing practical upper bounds on the Lipschitz constant of the classifier, for which a lower value would provide stronger formal guarantees of privacy. Further research into employing Lipschitz regularizations may both improve the performance of our method and provide stronger guarantees. In practice, we observe stronger protections than the formal guarantees provide. Thus, we include the formal guarantees primarily as a source of intuition.

## 6 CONCLUSION

We have presented a new approach to prevent memorized images from appearing during inference time. Our method is applicable to both guided and unguided diffusion models. Unlike previous methods intended to protect privacy of unguided diffusion models, CPSample does not necessitate retraining the denoiser. Moreover, the presence of duplicated data in the training corpus does not affect on our approach, and after training the classifier, one can adjust the level of protection enforced by CPSample without further training. We have shown theoretically that our method behaves similarly to rejection sampling without necessitating resampling. Finally, we have provided empirical evidence with rigorous statistical testing that our method is effective in unguided settings. We have also given examples in which CPSample was able to prevent extreme instances of mode collapse in Stable Diffusion. Despite its efficacy at preventing replication of training images, CPSample has little negative impact on image quality.

### REPRODUCIBILITY STATEMENT

We provide source code, scripts, and configuration details for experiments in the supplementary material for those seeking to reproduce this study. Proofs of original claims are given in the appendix, along with details for the implementation and training of the model. Statistical measures of significance are included to ensure the robustness of our results.

### ETHICS STATEMENT

This paper addresses privacy concerns in generative AI models. Our method is designed to mitigate these risks by safe guarding against membership inference attacks and data replication. We caution users that although our method prevents exact replication of training data, the theoretical assumptions are relaxed in practice, so it does not guarantee complete protection. We encourage others to act responsibly in the application of this work and to further research methods to prevent exact replication of trainign data.

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

## A  PROOFS

**Details of classifier guidance**    For completeness, we include a derivation of the classifier-guidance introduced in (Dhariwal & Nichol, 2021).

During the conditional denoising process, one should sample $x_{t-1}$ from the conditional distribution

$$\mathbb{P}(x_{t-1} \mid x_t, y) = \frac{\mathbb{P}(x_{t-1}, x_t, y)}{\mathbb{P}(x_t, y)} = \frac{\mathbb{P}(x_{t-1} \mid x_t)\mathbb{P}(y \mid x_t, x_{t-1})}{\mathbb{P}(y \mid x_t)}. \tag{A.1}$$

One can show that $\mathbb{P}(y \mid x_t, x_{t-1}) = \mathbb{P}(y \mid x_{t-1})$ (see Dhariwal & Nichol (2021) for details). The denominator $\mathbb{P}(y \mid x_t)$ does not depend on $x_{t-1}$. Therefore, we write this term as $Z$. To get an estimate of the probability $\mathbb{P}(y \mid x_{t-1})$, we train a classifier of the form $p_\phi(y \mid x_{t-1})$. Thus, we should estimate the conditional probability $\mathbb{P}(x_{t-1} \mid x_t, y)$ via

$$p_{\theta,\phi}(x_{t-1}, x_t, y) = Zp_\theta(x_{t-1} \mid x_t)p_\phi(y \mid x_{t-1}). \tag{A.2}$$

In continuous time, we can write $p(x_t, y) = p(x_t)p(y \mid x_t)$, and the score function is:

$$\nabla_{x_t} \log(p_\theta(x_t)p_\phi(y \mid x_t)) = \nabla_{x_t} \log p_\theta(x_t) + \nabla_{x_t} \log p_\phi(y \mid x_t). \tag{A.3}$$

The network $\epsilon_\theta(x_t, t)$ predicts the noise added to a sample, which can be used to derive the score function

$$\nabla_{x_t} \log p_\theta(x_t, t) = -\frac{1}{\sqrt{1 - \overline{\alpha}_t}}\epsilon_\theta(x_t, t).$$

Substituting this into equation A.3, we get

$$-\frac{1}{\sqrt{1 - \overline{\alpha}_t}}\epsilon_\theta(x_t) + \nabla_{x_t} \log p_\phi(y \mid x_t). \tag{A.4}$$

This leads to a new prediction for

$$\hat{\epsilon}_\theta(x_t) = \epsilon_\theta(x_t) - \sqrt{1 - \overline{\alpha}_t}\nabla_{x_t} \log p_\phi(y \mid x_t).$$

The conditional sampling then follows in the same manner as standard DDIM with $\epsilon_\theta$ replaced by $\hat{\epsilon}_\theta$.

**Proof of Lemma 1**

*Proof.* Let $x' \in B_\delta(x_0)$, where $x_0 \in D$ is assigned the random label $y$. By Lipschitz continuity, we have that

$$|p_\phi(y \mid x_0, t) - p_\phi(y \mid x', t)| < Ld(x_0, x').$$

By Assumption 2, we have $p_\phi(y \mid x_0, 0) > 1 - \kappa$, it follows that

$$\begin{aligned}
p_\phi(y \mid x', 0) &= p_\phi(y \mid x_0, 0) - p_\phi(y \mid x_0, 0) + p_\phi(y \mid x', 0) \\
&= p_\phi(y \mid x_0, 0) - (p_\phi(y \mid x_0, 0) - p_\phi(y \mid x', 0)) \\
&\geq p_\phi(y \mid x_0, 0) - |p_\phi(y \mid x_0, 0) - p_\phi(y \mid x', 0)| \\
&\geq p_\phi(y \mid x_0, 0) - Ld(x_0, x') \\
&\geq p_\phi(y \mid x_0, 0) - L\delta \\
&\geq 1 - \kappa - L\delta \\
&= 1 - \lambda.
\end{aligned}$$

Therefore, for all points $x' \in S$, we have $p_\phi(y \mid x', 0) \in [0, \lambda] \bigcup [1 - \lambda, 1]$. By Assumption 3, CPSample generates samples $\tilde{x}$ with $p_\phi(y \mid \tilde{x}) \in [\lambda, 1 - \lambda]$ with probability at least $1 - \varepsilon$. Thus, we have that CPSample generates samples outside of $S$ with probability at least $1 - \varepsilon$.    $\square$

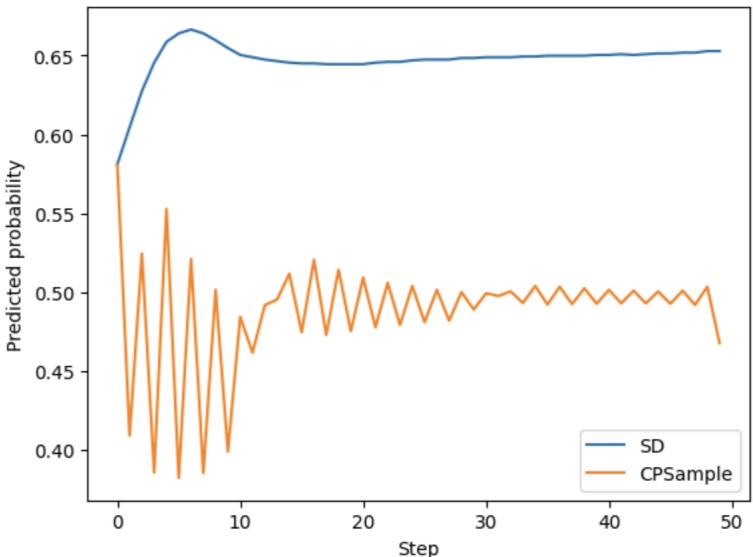

Figure 7: CPSample is able to generate images with $p_\phi(y \mid \tilde{x}, 0) \in (\lambda, 1 - \lambda)$. This example shows the probability $p_\phi(y = 1 \mid x_t, t)$ during the generation process with Stable Diffusion guided by the caption "Rambo 5 and Rocky Spin-Off - Sylvester Stallone gibt Updates." Note that a higher step indicates a later point in the denoising process. In this example, Stable diffusion exactly replicated the memorized image of Stallone, whereas CPSample ($\alpha = 0.5, s = 2000$) produced an original image.

# B  CLASS GUIDED DIFFUSION

As a final experiment, we implement CPSample alongside classifier-free guidance for CIFAR-10 to ensure that CPSample does not cause frequent out-of-category samples. The models used for guided diffusion were smaller, so the image quality is naturally lower.

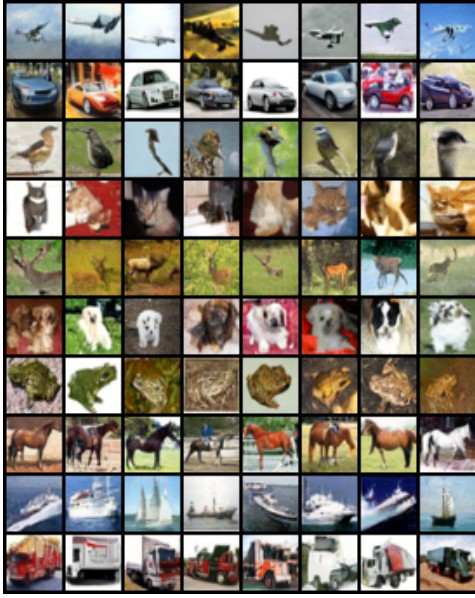

Figure 8: Uncurated samples using classifier-free guidance on CIFAR-10. The image in the position second row, third column from the top left is a near-exact replica of a member of the training data.

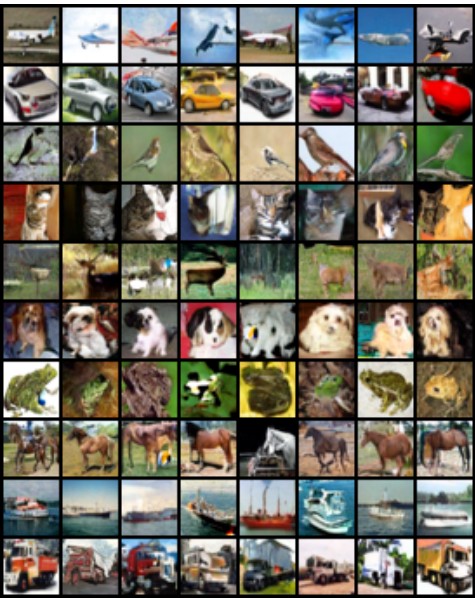

Figure 9: Uncurated samples using CPSample ($\epsilon = 0.1, s = 10$) along with classifier-free guidance on CIFAR-10. Note that although CPSample slightly reduces image quality, it does not cause out-of-category samples.

## C    TRAINING DETAILS

**Training classifiers.**    For training the classifier, we randomly selected subsets of $1\,000$ images each from the CIFAR-10, CelebA, and LSUN Church datasets, on which we trained the classifier from scratch. The architecture of our classifier is a modified version of the U-Net model. We retained key components of the U-Net (Ronneberger et al., 2015) model structure, including the timestep embedding, multiple convolutional layers for downsampling, and middle blocks. The output from the middle blocks underwent processing through Group Normalization, SiLU (Elfwing et al., 2017) activation layers, and pooling layers before being fed into a single convolutional layer, yielding the classifier's output. Parameters for layers identical to the standard U-Net were consistent with those used to pretrain the DDIM model on these datasets. Additionally, akin to the pretraining of DDIM, we incorporated Exponential Moving Average during training to stabilize the training process. The training of each classifier model was conducted using 4 NVIDIA A4000 GPUs with 16GB of memory. For subsets of $1\,000$ images, the classifier took only hours to train. For larger datasets consisting of $60\,000 - 160\,000$ data points, the classifier took up to 1 week to train. By comparison, retraining a diffusion model to be differentially private or using the method presented in (Daras et al., 2024) can take weeks or months depending on the dataset.

**Fine-tuning pretrained denoiser model on subsets.**    For fine-tuning the pretrained denoiser model on subsets, we commenced with the $500\,000$-step pretrained checkpoints available for the denoiser DDIM model. Fine-tuning was performed on subsets of $1\,000$ images each from the CIFAR-10, CelebA, and LSUN Church datasets until the model began generating data highly resembling the respective subsets. The number of training steps varied across different models, and specific details regarding the fine-tuning process can be found in Table 5. Throughout the fine-tuning process, hyperparameters remained consistent with those used during the pretraining phase. We employed 2 NVIDIA A5000 GPUs with 24GB of memory for fine-tuning each model on the subsets.

## D    EVALUATION DETAILS

**Numerical stability**    For the purposes of numerical stability, we slightly modified the sampling process described in Section 3.1. We noticed in earlier iterations of our method that very small numbers of images were becoming discolored or black because in float16, the classifier was predicting

Table 5: Training Parameters & Steps

|  | Batch Size | LR | Optimizer | EMA Rate | Classifier Steps | Fine-tune Steps |
|---|---|---|---|---|---|---|
| **CIFAR-10** | 256 | 2e-4 | Adam | 0.9999 | 560 000 | 110 000 |
| **CelebA** | 128 | 2e-4 | Adam | 0.9999 | 610 000 | 150 000 |
| **LSUN Church** | 8 | 2e-5 | Adam | 0.999 | 1 250 000 | 880 000 |

probabilities of 0.0000 or 1.0000 for the random label 1, causing the logarithm to blow up. To fix this in practice, we do the following. Sample $x_T \sim \mathcal{N}(0, I)$. For $t \in \{T, ...., 1\}$, if $p_\phi(y = 0|x_t, t) < \alpha$, replace $\epsilon_\theta(x_t, t)$ with

$$\hat{\epsilon}_{\theta,\phi}(x_t, t) = \epsilon_\theta(x_t, t) - s\sqrt{1 - \overline{\alpha}_t} \cdot \nabla \log(\tau + p_\phi(y = 0|x_t, t)).$$

If $p_\phi(y = 1|x_t, t) < \alpha$, replace $\epsilon_\theta$ with

$$\hat{\epsilon}_{\theta,\phi}(x_t, t) = \epsilon_\theta(x_t, t) - s\sqrt{1 - \overline{\alpha}_t} \cdot \nabla \log(\tau + p_\phi(y = 1|x_t, t)).$$

Otherwise, leave the sampling process unchanged.

By setting $\tau$ equal to 0.001, we were able to prevent the undesirable behavior.

**Similarity Reduction Evaluation.** We employ the fine-tuned denoiser model to generate 3 000 image samples for each of the aforementioned datasets. Additionally, we utilize the Classifier-guided method to generate another set of 3 000 images. Subsequently, we employ DINO (Caron et al., 2021) to find nearest neighbors in the subset using a methodology akin to ambient diffusion. From the perspectives of both DINO's similarity scores and human evaluation, we observe that images generated through the classifier-guided approach exhibit significantly lower similarity to the original images in the subset compared to those generated without guidance.

**FID Evaluation.** For each dataset, we utilize the denoiser model fine-tuned on the subset to generate 30 000 images under the guidance of the classifier. Subsequently, we employ the FID score implementation from the EDM (Karras et al., 2022) paper to compute the FID score.

**Inference Speed** Although speed was not a goal of our method, we provide some context for how fast it is compared to standard diffusion. We do our comparison using a batch size of 1 to generate 10 images with 50 denoising steps. CPSample with $\alpha = 0.5$ (i.e. computing gradients of the classifier at every step) had an average per-image generation time of $26.1 \pm 0.029s$. By contrast, standard stable diffusion had an average generation time of $23.92 \pm 0.055s$. Therefore, when the classifier is small compared to the size of the diffusion model, the added time cost is insignificant.

## E MEMBERSHIP INFERENCE ATTACKS

---

**Algorithm 1** Test statistic for membership inference attack against diffusion models (Matsumoto et al., 2023a)

---

**Input:** Target samples $x_1, ..., x_m$, CPSample denoiser $\hat{\epsilon}_{\theta,\phi}$, noise schedule $\overline{\alpha}_t = \prod_{s=1}^{t}(1 - \beta_s)$
total_error $\leftarrow$ 0
**for** $x$ in $\{x_1, ..., x_m\}$ **do**
    total_error $\leftarrow$ total_error $+ \|\epsilon - \hat{\epsilon}_{\theta,\phi}(\sqrt{\overline{\alpha}_t}x + \sqrt{1 - \overline{\alpha}_t}\epsilon, t)\|^2$
**end for**
mean_error $\leftarrow$ total_error$/m$.

---

In keeping with our goal of preventing membership inference attacks that are based on high similarity to a single member of the training set, we also perform a permutation test to ensure that we are not producing images that are anomalously close to the training data. Explicitly, we test the null hypothesis: generating images from CPSample produces images that are no more similar to the training data than they are to arbitrary points drawn from the data distribution. Our tests are performed in the same setting used in Section 4.1. Let $S = \{x_1, ..., x_k\}$ be the data used for fine-tuning. Let

$T = \{x_1, ..., x_k, x_{k+1}, ..., x_n\}$ be the entire training set. Finally, let $P = \{\tilde{x}_1, ..., \tilde{x}_m\}$ be samples from CPSample. Then our permutation test is as follows:

1. Sample $\tilde{x}_1, ..., \tilde{x}_k$ from $P$ without replacement. For each $\tilde{x}_i$, compute the quantity in 2.9 where the nearest neighbor is chosen among $S$. Let the similarity score of the most similar pair be $a$.

2. Repeat the following process $\ell$ times: sample $S^i \subset T$ without replacement from $T$ so that $|S^i| = k$. Sample $P^i$ without replacement from $P$ so that $|P^i| = k$. Compute the most similar image in $S^i$ for each member of $P^i$. Call the similarity of the most similar pair $a_i$.

3. For a pre-specified level $\alpha$, reject the null hypothesis if $\frac{1}{\ell}\sum_{i=1}^{\ell} \mathbf{1}\{a_0 > a_i\} > \alpha$.

The results can be found in Table 6. Note that the test fails to reject on CIFAR-10 and LSUN Church, but succeeds on CelebA. This is likely because we fine-tuned the CelebA model more extensively than the other two.

Table 6: Reduction in cosine similarity between generated images and nearest neighbor in fine-tuning data.

| Dataset | FT Steps | $\alpha$ | Scale | DDIM | CPSample |
|---|---|---|---|---|---|
| CIFAR-10 | 150k | 0.001 | 1 | 0.92 | 0.47 |
| CelebA | 650k | 0.001 | 1 000 | 0.99 | 0.99 |
| LSUN Church | 455k | 0.1 | 10 | 0.99 | 0.60 |

[1] $p$-values were computed using a $\chi^2$ test for $H_0$: CPSample did not reduce the fraction of images with similarity score exceeding the threshold.

## F  ADDITIONAL EMPIRICAL RESULTS

Table 7: FID score *w.r.t.* $\alpha$ and Scale on CIFAR-10.

| | $\alpha = 0.001$ | $\alpha = 0.01$ | $\alpha = 0.1$ | $\alpha = 0.25$ | $\alpha = 0.49$ |
|---|---|---|---|---|---|
| Scale $= 1$ | 4.14275 | 4.15467 | 4.19058 | 4.19208 | 4.21859 |
| Scale $= 5$ | 4.15772 | 4.20731 | 4.36005 | 4.58839 | 4.9566 |
| Scale $= 10$ | 4.18083 | 4.26594 | 5.05858 | 6.17326 | 7.88949 |
| Scale $= 100$ | 4.96727 | 16.7173 | 74.7247 | 113.199 | 139.626 |

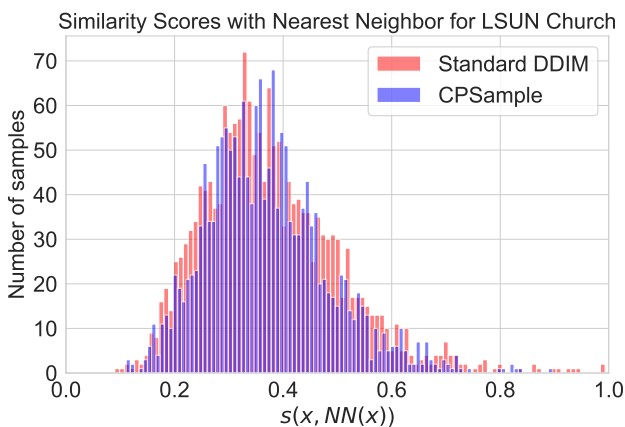

Figure 10: Similarity scores with nearest neighbor for standard DDIM and CPSample ($\alpha = 0.1$, scale$= 10$) on LSUN Church. In both cases, the network was fine-tuned for 455k gradient steps on a subset of 1 000 images.

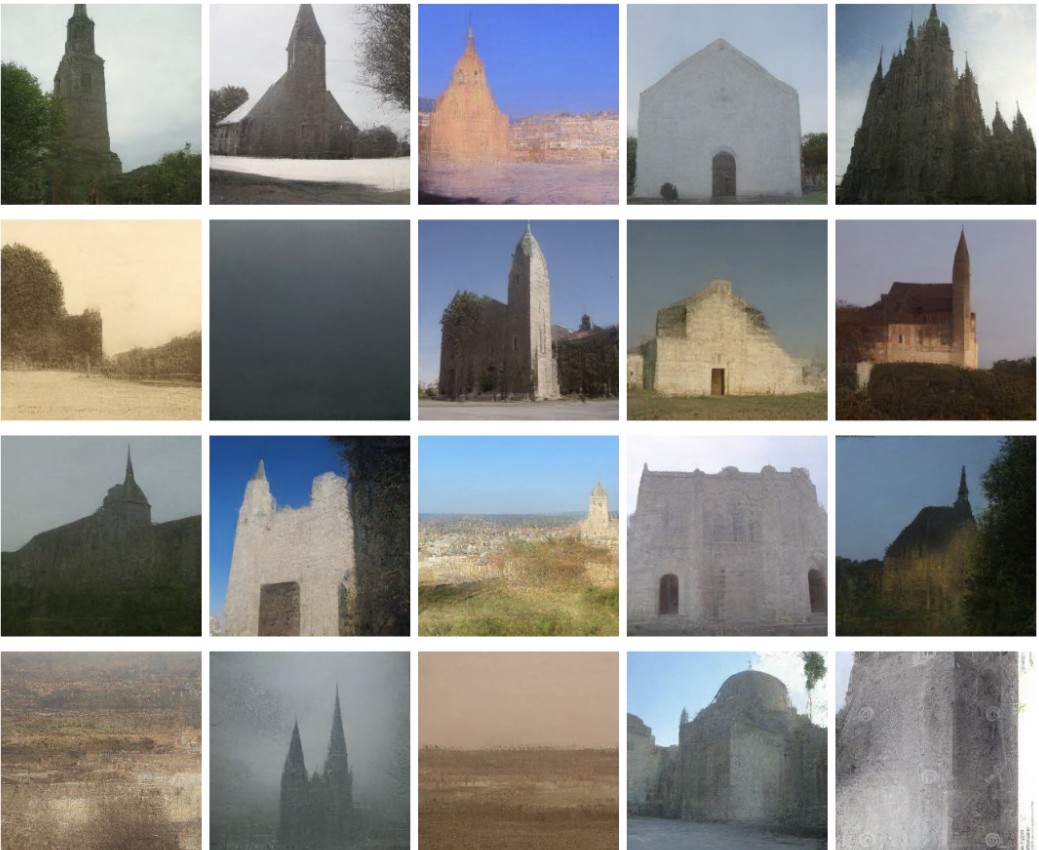

Figure 11: Uncurated samples using standard DDIM fine-tuned for 455k gradient steps on a subset of 1 000 images from LSUN Church.

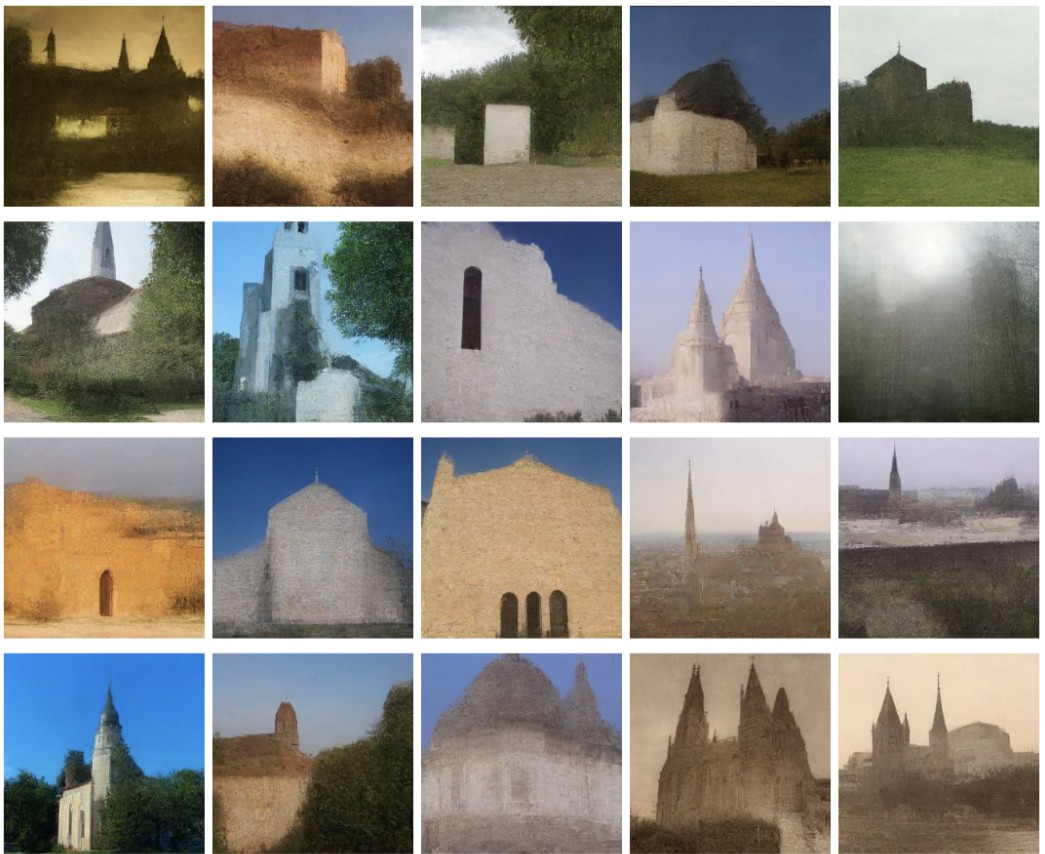

Figure 12: Uncurated samples using CPSample ($\alpha = 0.1$, scale= 10) applied to a network fine-tuned for 455k gradient steps on a subset of 1 000 images from LSUN Church. Note that there is no visual discrepancy in quality between these and the images from standard DDIM.

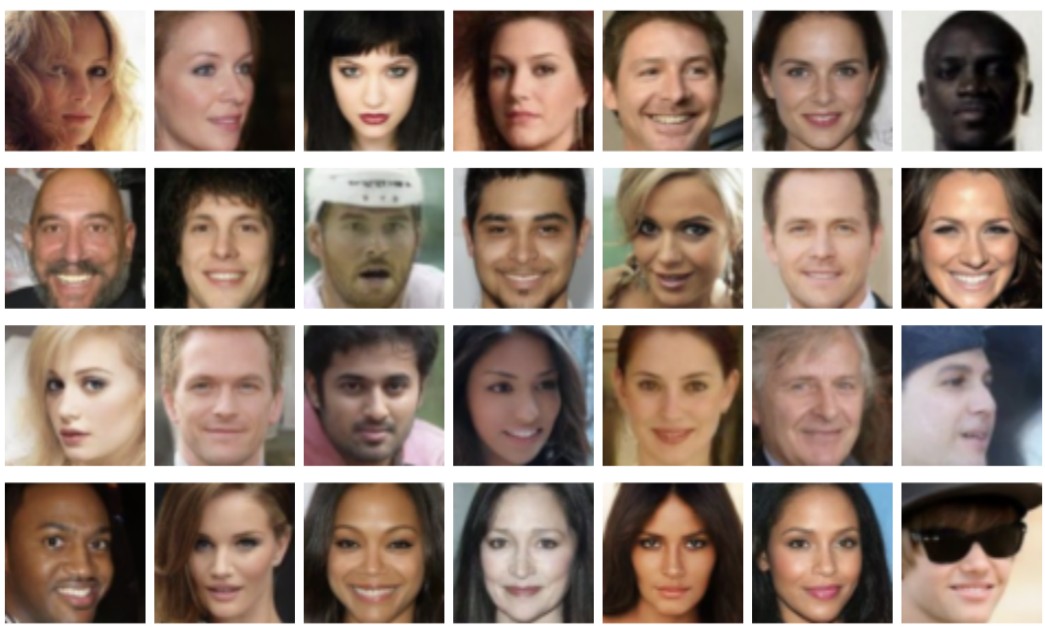

Figure 13: Uncurated samples using standard DDIM fine-tuned for 580k gradient steps on a subset of 1 000 images from CelebA.

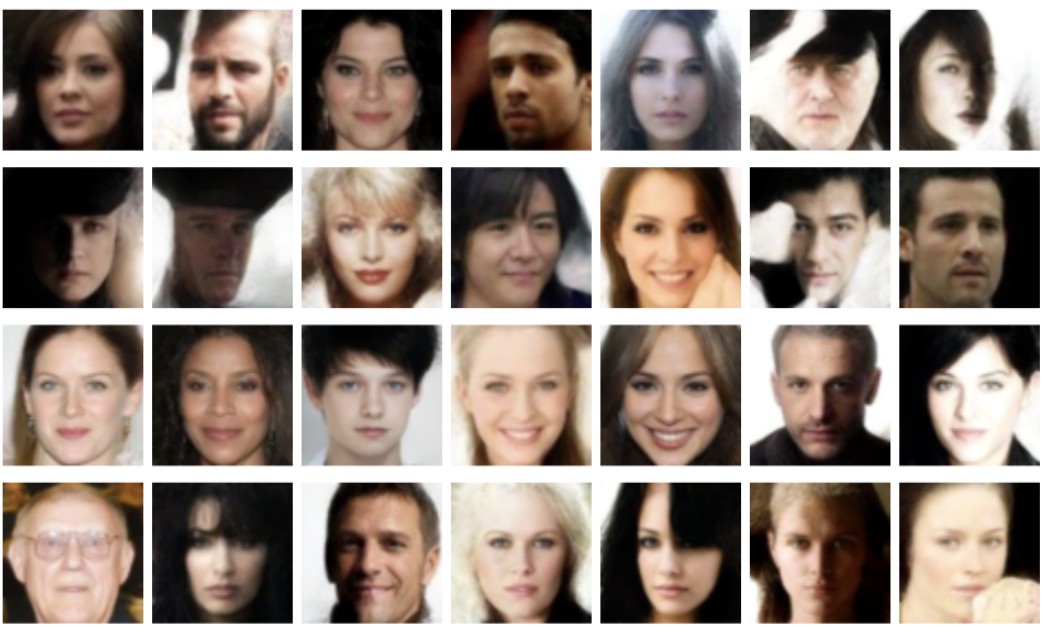

Figure 14: Uncurated samples using CPSample ($\alpha = 0.001$, scale$= 1\,000$) applied to a network fine-tuned for 580k gradient steps on a subset of 1 000 images from CelebA. Note that there is little visual discrepancy in quality between these and the images from standard DDIM.

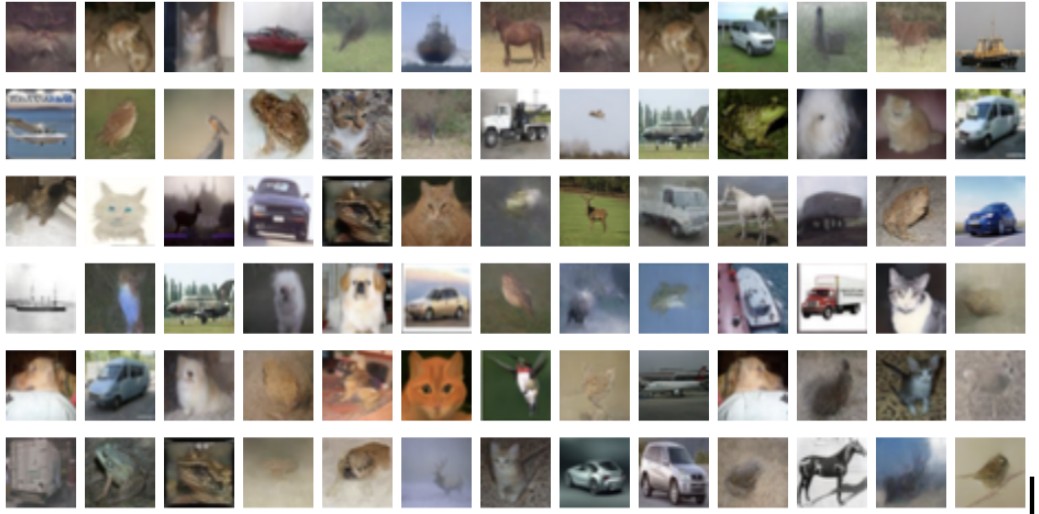

Figure 15: Uncurated samples using standard DDIM fine-tuned for 150k gradient steps on a subset of 1 000 images from CIFAR-10.

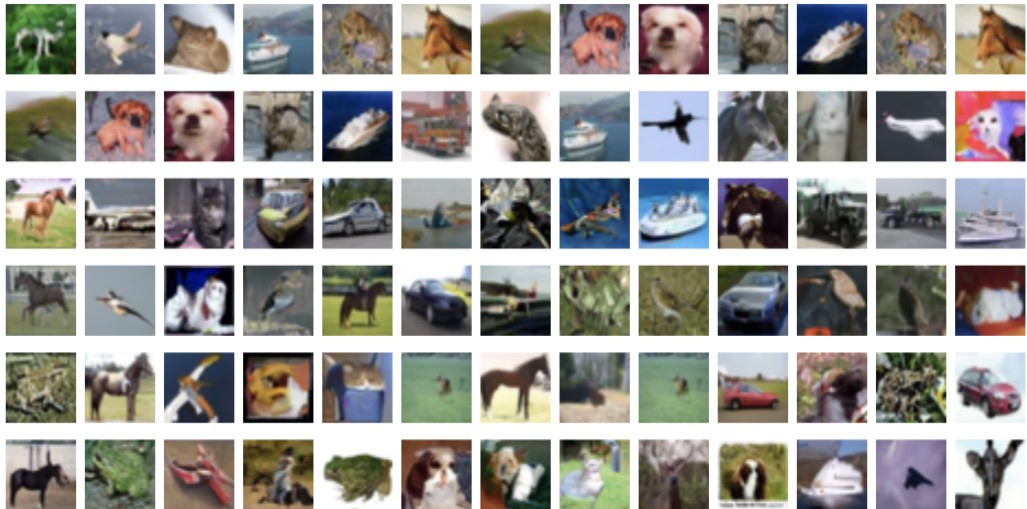

Figure 16: Uncurated samples using CPSample ($\alpha = 0.001$, scale= 1) applied to a network fine-tuned for approximately 150k gradient steps on a subset of 1 000 images from CelebA. Note that there is little visual discrepancy in quality between these and the images from standard DDIM.

