# OpenReview forum: "CPSample: Classifier Protected Sampling for Guarding Training Data During Diffusion"
_ICLR.cc/2025/Conference — ICLR 2025 Poster_

### Official Review · Reviewer_dYdC · 2024-10-20

**Soundness:** 3
**Presentation:** 2
**Contribution:** 2
**Rating:** 5
**Confidence:** 4

**Summary:**

This paper proposes a rejection sampling method that avoids diffusion models from replicating training data in generated contents. The main idea is to use classifier guidance dependent  on the label of training data / hold-out data. This gets the sampling process away from the region of training data and then preventing diffusion models from generating training data.

**Strengths:**

1) The method is sound and solid. It implements the idea of rejection sampling by using classifier guidance. This achieves training-free prevention of training data replication.

2) Experiments show that the proposed method can avoid training data replication even if there training setup leads to potential over-fitting. This proves the effectiveness of the proposed method,

**Weaknesses:**

1) Motivation: the main weakness of this paper lies in its motivation. As shown in existing research, training data replication is not a common phenomena in real-world pre-trained diffusion models. Most training samples are even unable to be inferred as member by membership inference attacks in Stable Diffusion, while the exception ones can be handled case by case. While the proposed method still degrades the generation performance as well as the diversity (potentially), it seems uneconomical to deploy the proposed method on any pre-trained diffusion models. Since privacy issues only happen in pre-trained diffusion models, the significance of the proposed method is moderate.

2) Computation cost & Generalizability: As stated in the paper, we need to train a classifier on all training and some hold-out examples. This could be significantly computationally costly without promising generalizability, since no one has trained such a classifier before. In fact, the training of the classifier in the paper only includes less than 2 thousands images. However, the training datasets of pre-trained diffusion models consist of billions of images. It leaves doubts whether a classifier can effectively give guidance based on these large-scale datasets.

3) OOD generation (minor): One potential drawback, as shown in Figure 4, is that the proposed method may generate images with contents quite different to the description of the text.

In general, although the proposed method is interesting and performs soundly on toy diffusion models, it does not seem to practical to deploy it on real-world AIGC applications which raises the data privacy concerns. Hence, I do not recommend accepting this paper.

**Questions:**

1) Can you show how your method performs on large-scale datasets?

---

> ### Author Response · Authors · 2024-11-17
> **Rebuttal to Reviewer dYdC**
>
> Thank you for reviewing our paper.  We're glad that you find our experiments convincing.  We address your concerns about our paper below:
>
> Weaknesses:
> >Motivation: the main weakness of this paper lies in its motivation. As shown in existing research, training data replication is not a common phenomena in real-world pre-trained diffusion models. Most training samples are even unable to be inferred as member by membership inference attacks in Stable Diffusion, while the exception ones can be handled case by case.
>
> To the best of our knowledge, memorization in diffusion models is a very real concern, and diffusion models are not robust to membership inference attacks [1].  Moreover, exact replication of training images is certainly a concern, as Carlini et. al. were able to extract many of the training images from Stable Diffusion [2].
>
> >While the proposed method still degrades the generation performance as well as the diversity (potentially), it seems uneconomical to deploy the proposed method on any pre-trained diffusion models.
>
> Compared to training differentially private models, which can be several times slower than training standard diffusion models, using CPSample is comparatively cheap, as one only needs to overfit a classifier.  Once trained, the classifier allows for flexible levels of protection without retraining.
>
> >Since privacy issues only happen in pre-trained diffusion models, the significance of the proposed method is moderate.
>
> We are not sure what the reviewer means by this comment.  Diffusion models are currently vulnerable to membership inference attacks.  Individuals training models for applications such as medical imaging still face the risk of leaking their training data when serving models that they have trained or fine-tuned on sensitive data.
>
> >Computation cost & Generalizability: [...] In fact, the training of the classifier in the paper only includes less than 2 thousands images.
>
> We trained classifiers on up to 160,000 images, as shown in Section 4.4.  As demonstrated by Section 4.3, the resulting models are less vulnerable to MI attacks, and Figure 5 shows that they produce high quality images without revealing the training data.  As stated in the paper, Zhang et. al. note that training a classifier on random versus real labels requires compute that only increases by a small constant factor as long as the number of network parameters is linearly proportional to the number of training data points [3].
>
> >OOD generation (minor): One potential drawback, as shown in Figure 4, is that the proposed method may generate images with contents quite different to the description of the text.
>
> The images seem to be faithful to their captions to us.  The first shows Sylvester Stallone, the second shows a classic car, and the third is abstract art.
>
> >Can you show how your method performs on large-scale datasets?
>
> Which experiments would you like to see?  We are an academic lab and cannot train from scratch on extremely large datasets.  We showed that CPSample can protect datasets of up to 160,000 images, but we are resource-limited and have not tried to train on larger datasets than that.  Moreover, if one needs to protect specific images (i.e. Sylvester Stallone) within a multi-billion image dataset, we have shown that CPSample is effective.
>
> >it does not seem to practical to deploy it on real-world AIGC applications which raises the data privacy concerns
>
> We deployed this method with Stable Diffusion to protect a subset of the LAION dataset.  Is there another experiment that you would like to see?
>
>
>
> [1] https://arxiv.org/pdf/2301.09956
> [2] https://arxiv.org/pdf/2301.13188
> [3] Chiyuan Zhang, Samy Bengio, Moritz Hardt, Benjamin Recht, and Oriol Vinyals. Understanding deep learning requires rethinking generalization. In 5th International Conference on Learning Representations, ICLR 2017, Toulon, France, April 24-26, 2017, Conference Track Proceedings. OpenReview.net, 2017. URL https://openreview.net/forum?id=Sy8gdB9xx.

---

> ### Comment · Reviewer_dYdC · 2024-11-21
>
> I agree that the memorization of diffusion models is a big deal. However, according to [1, 2], only few proportions of images can be detected as memorization. For example, in [1], searching from 350,000 most duplicated images in Stable Diffusion only gets 94 of them as memorized images. I believe some post-training checks and strategies can cheaply avoid the model from generating these images. In other words, a practical way to solve the memorization problem should be, before releasing the model, conducting a memorization check, finding the memorized images, and doing some post-training refinements specifically for these images. This will not add any extra computational costs to the inference as the proposed method. The proposed method not only asks for extra costs in inference (users will not like this) but also requires a mounted classifier which could be easily bypassed (as the users of Stable Diffusion do to the safety checker). Hence, I think this will not be a practical method to solve the real-world memorization problem.

---

### Official Review · Reviewer_t8KP · 2024-10-30

**Soundness:** 3
**Presentation:** 2
**Contribution:** 3
**Rating:** 5
**Confidence:** 4

**Summary:**

Diffusion models exhibit a tendency to memorize training images and reproduce these training images during inference, which can pose significant privacy concerns. To address this issue, this paper introduces a sampling strategy (CPSample) guided by a trained classifier. This classifier evaluates the memorization score of current generated image and rectifies the sampling trajectories to diverge from the training images. By adjusting the guidance scale, the proposed method ensures that the generated images possess a memorization score approximating 0.5, thereby mitigating the risk of replicate generation.

**Strengths:**

1. The issue of replicate generation in diffusion models is critical, and there is an urgent need for a scalable solution to address it.
2. The concept of explicitly training a classifier to measure the memorization score is novel, and subsequent experiments have demonstrated its efficacy.

**Weaknesses:**

1. The scalability of the strategy to train a classifier is a primary concern. Given that current text-to-image diffusion models, such as Stable Diffusion and Pixart, are trained on large-scale datasets scraped from the Internet, is it a feasible solution to train an overfitted classifier on a large scale dataset?

2. The proposed method assumes that the training member set is available. However, this assumption does not always hold. Specifically, the training member sets of certrain text-to-image diffusion models, such as Pixart, are not publicly accessible, therby limiting the applicability of the proposed approach.

3. While the proposed CPSample method effectively mitigates replicate generation, it also exhibits side effect on the generated images as it manipulates the sampling trajectory. For example, in Figure 4, the generated image by CPSample does not align with the input prompt and sometimes will generate corrupted results. In contrast, although rejection sampling faces challenges with resampling, it tends to produce images that are more closely aligned with the input prompt.

4. There are many typos in notations of DDPM, for example, $\alpha_t \rightarrow \bar{\alpha}_t$ in Equation 2.1&2.2, Line 146. Please check this through the whole manuscripts to ensure the consistency and accuracy of all notations.

**Questions:**

1. In my opinion, the trained overfitted classifier is designed to estimate a memorization score and captures a boundary between the member set and hold-out set to provide a guided gradient. However, various alternative methods exist for estimating this boundary or providing a similar memorization gradient. For instance, training a classifier to differentiate between the member set and hold-out set, or training an overfitted diffusion model to provide memorization gradients, are both plausible approaches. Therefore, why choose to assign random 0/1 labels to the member set? (The reasons provided in Line 238-244 is not very convincing for me)

2. The scalability of CPSample to diffusion models with larger member sets is uncertain. Current experiments only provide limited examples (Section 4.2). Additional examples are required to further demonstrate the effectiveness of the proposed method. Furthermore, the hyperparameters ($\alpha$, $s$) for CPSample seems dependent on each sampling. How to define these hyperparameters for each sampling?

3. The experiments on similarity reduction have been conducted on low-resolution and small-scale datasets such as Cifar10 and CelebA. For these datasets, the diffusion model can verbatim memorize each pixel of the training samples. However, in more practical scenarios, diffusion models tend to memorize only the rough structure of the training sample and can only reproduce similar patterns. Consequently, the current experiments (Section 4.1) may not adequately demonstrate the effectiveness of the proposed method for similarity reduction. Additional experiments on large-scale diffusion models are necessary to further validate the method.

4. The current experiments have been mostly conducted on unconditional diffusion models. To fully evaluate the effectiveness of CPSample, additional experiments on conditional diffusion models, such as text-to-image diffusion models, should be conducted to assess whether CPSample can accurately align with the given condition.

---

> ### Author Response · Authors · 2024-11-18
> **Rebuttal to Reviewer t8KP**
>
> Thank you for reviewing our paper.  We're glad that you share our view of the problem's importance.  We address some of your concerns about our work below:
>
> ## Weaknesses:
> >The scalability of the strategy to train a classifier is a primary concern. Given that current text-to-image diffusion models, such as Stable Diffusion and Pixart, are trained on large-scale datasets scraped from the Internet, is it a feasible solution to train an overfitted classifier on a large scale dataset?
>
> We trained on up to 160,000 images without encountering problems.  Zhang et. al. [1] show that training a classifier on random versus real labels requires compute that only increases by a small constant factor as long as the number of network parameters is linearly proportional to the number of training data points, suggesting that the method should be scalable.
>
> >The proposed method assumes that the training member set is available. However, this assumption does not always hold. Specifically, the training member sets of certrain text-to-image diffusion models, such as Pixart, are not publicly accessible, therby limiting the applicability of the proposed approach.
>
> If one no longer has the original training data available, one can identify images that are at risk of being duplicated during training (for instance, using Carlini et. al. [2]) and train the classifier on those.
>
> >For example, in Figure 4, the generated image by CPSample does not align with the input prompt and sometimes will generate corrupted results. In contrast, although rejection sampling faces challenges with resampling, it tends to produce images that are more closely aligned with the input prompt.
>
> While there are side effects to CPSample, they are minor compared to alternatives such as DPDM or ambient diffusion in terms of quality, as we show in Section 4.4.  We argue that the images in Figure 4 are faithful to their prompts-- the third caption requested abstract art, so we don't think that this is an example of a corrupted image.
>
> >There are many typos in notations of DDPM
>
> Thank you for pointing this out.  We will carefully revise the paper.
>
> ## Questions:
> >In my opinion, the trained overfitted classifier is designed to estimate a memorization score and captures a boundary between the member set and hold-out set to provide a guided gradient. However, various alternative methods exist for estimating this boundary or providing a similar memorization gradient. For instance, training a classifier to differentiate between the member set and hold-out set, or training an overfitted diffusion model to provide memorization gradients, are both plausible approaches. Therefore, why choose to assign random 0/1 labels to the member set? (The reasons provided in Line 238-244 is not very convincing for me)
>
> We explicitly sought to avoid retraining a diffusion model, since this is extremely expensive, especially for large datasets.  Training a classifier on random 0-1 labels is much cheaper than training a diffusion model.  To clarify, the 0-1 labels were assigned to two halves of the training data, not to the validation data.  This allowed us to provide classifier guidance during samples away from the closest examples and towards others.  This prevented us from moving too far away from the data as a whole while still avoiding close proximity to any one data point.  Please let us know if you have other questions about this aspect of the paper.
>
> >The scalability of CPSample to diffusion models with larger member sets is uncertain
>
> We addressed this in the weaknesses section.
>
> >in more practical scenarios, diffusion models tend to memorize only the rough structure of the training sample and can only reproduce similar patterns.
>
> As shown in Figure 4, there are many examples where diffusion models replicate or nearly replicate the entire image.  CPSample is mainly focused on preventing this type of replication.  Replicating only the general structure of an image is less likely to be a severe privacy concern.
>
> >The current experiments have been mostly conducted on unconditional diffusion models. To fully evaluate the effectiveness of CPSample, additional experiments on conditional diffusion models, such as text-to-image diffusion models, should be conducted to assess whether CPSample can accurately align with the given condition.
>
> We provide two experiments on conditional diffusion models, one in Figure 4 and one in Appendix B.  Which additional experiments would you like to see (subject to reasonable compute constraints)?
>
>
>
> [1] Chiyuan Zhang, Samy Bengio, Moritz Hardt, Benjamin Recht, and Oriol Vinyals. Understanding deep learning requires rethinking generalization. In 5th International Conference on Learning Representations, ICLR 2017, Toulon, France, April 24-26, 2017, Conference Track Proceedings. OpenReview.net, 2017. URL https://openreview.net/forum?id=Sy8gdB9xx.
> [2] https://arxiv.org/abs/2301.13188

---

> > ### Comment · Reviewer_t8KP · 2024-11-19
> >
> > >Scalability
> >
> > I do not agree that 160,000 images are enough to prove the scalability of the proposed method. [1] is published in 2017 and only conducts experiments on low-resolution small-scaled datasets. However, data duplication is another story which occurs in diffusion models trained on high-resolution large-scale datasets. I do not agree the parameter scale linearly with the increase in resolution and the size of the dataset. If such linearity were to hold, the scaling law in the LLMs would also be linear. Consequently, I maintain my believe that there exists a limit to train an overfitted classifier on extremely large dataset, such as LAION-400M or LAION-5B. It is crucial that the authors provide more experiments to show the limitations of this method.
> >
> > >Accessibility of the training members
> >
> > I suggest that the authors give a more detailed practical scenarios in the introduction and add an limitation section to elaborate this.
> >
> > >what to approximate the boundary
> >
> >  Maybe my comments are somewhat misleading. What I want to emphasize is that the core design of this paper is the classifier, which is utilized to captures the boundary. However, no experiments are provided about this given that there are many plausible approaches to captures this boundary. Experiments on this will give a firm validation on choosing classifier as the solution.
> >
> > >structure replication
> >
> > I do not agree that structure replication is not a severe privacy concern. For example, duplications in IP or the art style are also of great significance.
> >
> > >additional experiments on conditional diffusion models
> >
> > For instance, class label to image models, edge to image models (controlnet). Besides, current qualitative results are very limited. More results should be provided.

---

### Official Review · Reviewer_8m3n · 2024-11-01

**Soundness:** 1
**Presentation:** 2
**Contribution:** 2
**Rating:** 3
**Confidence:** 4

**Summary:**

The paper proposes a method to prevent the generation of training data during the inference on diffusion models. The way to obtain the result is by modifying the sampling process in diffusion models. The method leverages a classifier that is trained and forced to overfit on random binary labels assigned to the training data of diffusion models. The method uses “classifier guidance” to prevent the generation process from returning the training data points.

**Strengths:**

1. The paper tries to solve a very important problem: how to prevent the diffusion models from generating the training samples.

**Weaknesses:**

1.	This method is very simple: “Overfit a classifier on random binary labels assigned to the training data and use this classifier during sampling to guide the images away from the training data.” Does it still require training the classifier on the large training dataset of diffusion models? What if we do not have access to the training data any longer?
2.	As stated in Section 2.1: “Currently, the state-of-the-art for guided generation is achieved by models with classifier-free guidance (Ho & Salimans, 2022). However, since CPSample employs a classifier to prevent replication of its training data,” Thus, the CPSample method is not applicable to the SoTA diffusion models.
3.	The proposed CPSample method is very similar to the AMG guidance method from Chen et al. 2024. It is claimed that AMG requires access to the training data of a diffusion model. How is the classifier trained for the proposed CPSample method without requiring access to the training data? (as stated at the end of Section 3.1)
4.	The qualitative results in Figure 3 clearly show the degradation of quality in the generated images between DDIM and CPSample (especially for the LSUN Church dataset). Additionally, Image C in the last row for the CPSample shows the complete degradation in the quality of the generated images.
5.	What is the overhead of the method in terms of the inference time? How long does it take to train the classifier? How is the classifier trained?
6.	The paper only compares the FID score of the CPSample to other methods, however, no comparison is presented in terms of the reduction in the generation of the training data points.

**Questions:**

1.	The abstract states: “CPSample achieves FID scores of 4.97 and 2.97 on CIFAR-10 and CelebA-64, respectively, without producing exact replicates of the training data.” What is the baseline? What model was it checked on? Without this background and some reference, these results do not have any meaning.
2.	The paper should cite another work that applied DP to diffusion models [1].
3.	Introduction: this method “fortifies against some membership inference attacks” This statement is rather too vague and it should be clearly stated what the goal of the method is. The defenses should not prevent some attacks that can infer the training images because the adversary would simply use the attacks that work and are not prevented by the proposed method.
4.	The fact that better FID scores are achieved than other methods has to be presented with some reference. Is it the trade-off that the higher performance stems simply from the worse protection?
5.	The background of the paper takes 2 pages. It is way too long and should be limited to a single page. For example, DP is not used in this paper so there is no need to provide this background section. Moreover, providing the formula for cosine similarity is really too trivial. The statement about the empirical results from this paper in Section 2.2 is very surprising: “Empirically, for CIFAR-10, we observed that images with similarity scores above 0.97 were nearly identical, whereas for CelebA, the threshold was approximately 0.95, and for LSUN Church, images with similarity above 0.90 were sometimes, though not always, nearly identical.” The same is repeated in Section 4.1.
6.	The definition of Differential privacy does not come from Dwork & Roth 2014. The first paper that proposed the mathematical framework was [2].
7.	Why is only a single membership inference attack used by Matsumoto et al. 2023a? The latest “evaluation of state-of-the-art MIAs on diffusion models reveals critical flaws and overly optimistic performance estimates in existing MIA evaluation” [3]
8.	Why are only the binary labels used and not more labels in the CPSample?
9.	From Section 4.1: Figure 1 does not show the most similar pairs of samples and fine-tuning data points.

**References:**

[1] dp-promise: Differentially Private Diffusion Probabilistic Models for Image Synthesis. Haichen Wang, Shuchao Pang, Zhigang Lu, Yihang Rao, Yongbin Zhou, Minhui Xue. USENIS 2024.

[2] Calibrating noise to sensitivity in private data analysis. Dwork, C., McSherry, F., Nissim, K., & Smith, A. In Theory of Cryptography: Third Theory of Cryptography Conference, TCC 2006.

[3] Real-World Benchmarks Make Membership Inference Attacks Fail on Diffusion Models. Chumeng Liang, Jiaxuan You 2024: https://arxiv.org/abs/2410.03640

---

> ### Author Response · Authors · 2024-11-19
> **Rebuttal to Reviewer 8m3n (Part 1)**
>
> Thank you for reviewing our paper and providing helpful suggestions.  We address the weaknesses and questions that you brought up below.
>
> ## Weaknesses
> >This method is very simple: “Overfit a classifier on random binary labels assigned to the training data and use this classifier during sampling to guide the images away from the training data.”
>
> Is methodological simplicity a weakness?
>
> >Does it still require training the classifier on the large training dataset of diffusion models? What if we do not have access to the training data any longer?
>
> As shown by Carlini et. al., vulnerable training data can be extracted from the diffusion model.  One can train the classifier to protect this vulnerable data if the original data is no longer available.  A more realistic use case is that one could fine-tune a pretrained model using one's own sensitive data and use CPSample to protect this sensitive data.
>
> >Thus, the CPSample method is not applicable to the SoTA diffusion models.
>
> The reviewer is mistaken.  As we show in Section 4.2, CPSample can protect models like Stable Diffusion that are not steered by a classifier.
>
> >The proposed CPSample method is very similar to the AMG guidance method from Chen et al. 2024.
>
> We fail to understand the connection between CPSample and AMG.  These seem to be methods used in different settings for different purposes. Could the reviewer please explain the connection?
>
> >How is the classifier trained for the proposed CPSample method without requiring access to the training data?
>
> You do require access to the training data or at least knowledge of which images you wish to protect from replication. This is the case for the methods that we compare against, too (ambient diffusion, DPDM).
>
> >The qualitative results in Figure 3 clearly show the degradation of quality in the generated images between DDIM and CPSample (especially for the LSUN Church dataset).
>
> These are not randomly chosen images-- they are the generated images that are most similar to the training data.  Hence, it is unsurprising that the images from the original diffusion model (which are identical to training images) are of higher quality.  The FID scores in Table 4 are a more fair metric of quality, and they show minimal degradation.  For randomly sampled images, which would make for a more fair visual comparison, see Appendix F.
>
> >Additionally, Image C in the last row for the CPSample shows the complete degradation in the quality of the generated images.
>
> The caption requested abstract art on several panels-- the image looks like abstract art on several panels.  Note that it is the only image that looks "abstract."
>
> >What is the overhead of the method in terms of the inference time? How long does it take to train the classifier? How is the classifier trained?
>
> Training details are provided in Appendix C.  Inference times are commensurate with those for classifier-guided diffusion models.  We will be happy to include timed trials in the final version of the paper if the reviewer recommends it.
>
> >The paper only compares the FID score of the CPSample to other methods, however, no comparison is presented in terms of the reduction in the generation of the training data points.
>
> In the FID score comparison, neither CPSample nor the methods that we compared against produced any duplicates of the training data.  Thus, we feel it is a fair comparison.  Is there another experiment that the reviewer feels would strengthen the comparison?

---

> > ### Author Response · Authors · 2024-11-19
> > **Rebuttal to Reviewer 8m3n (Part 2)**
> >
> > ## Questions
> >
> > >The abstract states: “CPSample achieves FID scores of 4.97 and 2.97 on CIFAR-10 and CelebA-64, respectively, without producing exact replicates of the training data.” What is the baseline? What model was it checked on? Without this background and some reference, these results do not have any meaning.
> >
> > We describe the model architectures in Appendix C and the evaluation details in Appendix D.  To restate, the baseline models were those from Song et. al.'s DDIM paper.  We applied CPSample directly to these models and checked that no images were duplicated from the training data before evaluating FID scores.  We are happy to answer any other questions that the reviewer has about our evaluations.
> >
> > >The paper should cite another work that applied DP to diffusion models [1].
> >
> > We will be happy to include this reference.
> >
> > >Introduction: this method “fortifies against some membership inference attacks” This statement is rather too vague and it should be clearly stated what the goal of the method is. The defenses should not prevent some attacks that can infer the training images because the adversary would simply use the attacks that work and are not prevented by the proposed method.
> >
> > The only known method that is guaranteed to prevent all MIAs is differential privacy.  As we discussed in the introduction, this is not a tenable solution because of its effect on quality.  Therefore, we checked to ensure that CPSample protects against the most popular MIA for diffusion models.
> >
> > >The background of the paper takes 2 pages. It is way too long and should be limited to a single page. For example, DP is not used in this paper so there is no need to provide this background section. Moreover, providing the formula for cosine similarity is really too trivial. The statement about the empirical results from this paper in Section 2.2 is very surprising: “Empirically, for CIFAR-10, we observed that images with similarity scores above 0.97 were nearly identical, whereas for CelebA, the threshold was approximately 0.95, and for LSUN Church, images with similarity above 0.90 were sometimes, though not always, nearly identical.” The same is repeated in Section 4.1.
> >
> > Thanks for pointing this out.  We will cut down the introductory material and remove repetitions.
> >
> > >The definition of Differential privacy does not come from Dwork & Roth 2014. The first paper that proposed the mathematical framework was [2].
> >
> > Thanks for noting this-- we will correct the citation.
> >
> > >Why is only a single membership inference attack used by Matsumoto et al. 2023a? The latest “evaluation of state-of-the-art MIAs on diffusion models reveals critical flaws and overly optimistic performance estimates in existing MIA evaluation” [3]
> >
> > We provide a second membership inference attack in Appendix E.  We will be happy to include more.  Which membership inference attacks do you find particularly compelling?
> >
> > >Why are only the binary labels used and not more labels in the CPSample?
> >
> > If there are too many labels, the classifier guidance could end up steering the generations towards a particular image (if the classes get smaller and smaller).  We found that it was sufficient to use 2 classes to prevent replication of the training data, so we did not see a need to add more.
> >
> > >From Section 4.1: Figure 1 does not show the most similar pairs of samples and fine-tuning data points.
> >
> > We fine-tuned for too many iterations on a subset of the CelebA dataset, causing extreme mode collapse.  Therefore, we show a reduction in the rate of replication rather than a total elimination of replication in Figure 1.  Full statistics about replication rates can be found in table 1.

---

> ### Comment · Reviewer_8m3n · 2024-11-25
>
> Thank you for your answer, I appreciate it.
>
> >**We will be happy to include timed trials in the final version of the paper if the reviewer recommends it.**
>
> ICLR allows authors to conduct additional experiments and even update their submissions during the review process. Therefore, the absence of the recommended experiments before the final decision significantly diminishes the paper's chances of acceptance.
>
> This is not only the case for my review, but for other reviewers as well. For example, Reviewer 6foL asked for additional experiments with other MIA methods (which I also think should be added), however, they were not provided. The same holds for Reviewers dYdC and t8KP.

---

### Official Review · Reviewer_6foL · 2024-11-04

**Soundness:** 3
**Presentation:** 3
**Contribution:** 3
**Rating:** 5
**Confidence:** 4

**Summary:**

The paper proposes Classifier-Protected Sampling (CPSample), a novel technique to prevent diffusion models from generating training data  duplicates, apart from previous methods such as rejection sampling or other sampling methods that occurs during training phase, CPSample allows the diffusion model to steer away from generating training samples during test time, resolving the problem of privacy leakage.

**Strengths:**

1. The proposed method presents interesting facet and novelty where a classifier is utilized in remembering the statistics of training distribution and acts as guidance to steer away from the training data.
2. The planned experiments are great in terms of studying the effect of CPSample in a holistic manner. The three different experiments demonstrate different aspect of CPSample which is helpful for understanding the empirical effect.

**Weaknesses:**

1. The presented theoretical study seems to be not proven much in terms of CPSample. Specifically, it is not demonstrated as to why assumption 3 holds well empirically as stated in the paper. Furthermore, given the intractable estimation and sometimes large values of Lipschitz constant, I wonder if that would make the possible $\delta$ extremely small which in terms renders the theory results in vain?
2. Some more experiments on MIA could be carried out to effectively evaluated the information loss caused when CPSample. Particularly, papers such as [1, 2] have devised new MIAs tailored to the settings of diffusion models which also conforms to the past settings in MIA.  It would be great to see if CPSample could also withstand these newly devised attacks.



[1]: Duan, Jinhao, et al. "Are diffusion models vulnerable to membership inference attacks?." International Conference on Machine Learning. PMLR, 2023.
[2]: Kong, Fei, et al. "An efficient membership inference attack for the diffusion model by proximal initialization." arXiv preprint arXiv:2305.18355 (2023).

**Questions:**

1. As mentioned in the first point of weaknesses, it would be great if the authors could elaborate more on the the third assumptions either empirically or theoretically along with the $\delta$ problem mentioned.

---

> ### Author Response · Authors · 2024-11-18
> **Rebuttal to Reviewer 6foL**
>
> Thank you for reviewing our paper and offering these helpful suggestions.  We address the weaknesses and questions that you brought up below:
>
> ## Weaknesses
> >The presented theoretical study seems to be not proven much in terms of CPSample. Specifically, it is not demonstrated as to why assumption 3 holds well empirically as stated in the paper. Furthermore, given the intractable estimation and sometimes large values of Lipschitz constant, I wonder if that would make the possible $\delta$ extremely small which in terms renders the theory results in vain?
>
> We would be happy to include a study of how frequently assumption 3 holds in practice.  We provide an illustration in Figure 6 of how this assumption holds in practice, but we agree that this is likely insufficient verification.  Are there particular experiments that you would like to see in this direction?  We intended the theory to be a heuristic for why CPSample works, but we are open to suggestions that would help turn it into a more practical guarantee.
>
> > Some more experiments on MIA could be carried out to effectively evaluated the information loss caused when CPSample. Particularly, papers such as [1, 2] have devised new MIAs tailored to the settings of diffusion models which also conforms to the past settings in MIA. It would be great to see if CPSample could also withstand these newly devised attacks.
>
> The membership inference attack given in Algorithm 1 and used in Section 4.3 is extremely similar to the attack in Duan et. al.  Both inference attacks use reconstruction error as the basis for distinguishing between members of the training and held-out sets.  We are happy to include more examples of membership inference attacks in the final version of the paper.
>
> ## Questions
> >As mentioned in the first point of weaknesses, it would be great if the authors could elaborate more on the the third assumptions either empirically or theoretically along with the  problem mentioned.
>
> Please see the Weaknesses section.

---

> > ### Comment · Reviewer_6foL · 2024-11-26
> > **Response to Rebuttal -- Reviewer 6foL**
> >
> > Thanks for the response provided in the rebuttal. I appreciate the authors' efforts to make the theory a practical guarantee. On the other hand, similar to the review provided by Reviewer 8m3n, I believe providing more MIA results with the suggested experimental results is necessary and, in the authors' words, of reasonable computing constraint. Since there are no such options for a major revision in ICLR, I believe the authors must provide such basic results during the rebuttal period.

---

### Meta-Review · Area_Chair_DVmr · 2024-12-20

**Metareview:**

This paper studies the problem of alleviating privacy leakage in diffusion models, which occurs when these reproduce samples very close to those in training data and membership inference attacks. This is an important problem, given that state-of-the-art models are often trained in private or copy-righted data. The authors' proposed approach consists of fitting a classifier on the samples to be protected with random binary labels, and then employing it to steer the diffusion process away from those samples at inference time. While empirical and not endowed with theoretical guarantees (like those in differential privacy), experiments show that the approach effectively diminishes the reproduction of training samples, while compromising the sampling quality only mildly.

**Strengths**
* The problem studied is very relevant to modern SOTA diffusion models.
* The presented approach is conceptually simple.
* Their approach does not require re-training the diffusion model, potentially applying to off-the-shelf models,

**Weaknesses**
* The motivation of the work, and conceptual advantages w.r.t. other methods, could be presented more clearly.
* The need to train a classifier to 'memorize' the protected data could be a significant computational overhead.

**Summary**
In all, this paper provides a potential solution to the problem of reproduction of training samples from diffusion models, including SOTA models like Stable Diffusion. This is a hard problem, and the method does a reasonable job at a new, simple, and innovative solution. Most reviewers agree that the approach is effective in accomplishing this task.
The computational overhead, and the need to have access to the data to protect, seems almost unavoidable, and it might be cost to pay for protecting the private data. A significant amount of discussion was entertained by the reviewers, who are mostly leaning to reject - see further comments below. However, upon carefully all comments, responses, and the paper, this AC thinks that the strengths outweigh the weaknesses of this paper, and the recommendations from the reviewers fail to recognize this. I am therefore recommending acceptance with the following recommendations:

* Motivation: most reviewers were concerned with the need to have access to training data. In the discussions, the authors stressed that access to all of the training data might not be needed, (just) access to the data to be protected (a subset) is needed. Please make this motivation clear, positioning this into a practical scenario (as argued in your responses).
* Computational overhead: explain how and why the computational and memory overhead of your method is comparable to other approaches (as argued in your responses).
* Improve the presentation by incorporating new references and comments, where appropriate (as suggested by reviewers).
* Theoretical basis: Please stress the limitations of the provided theoretical results, as well as the limitation of this method in providing theoretical guarantees against privacy leakage (unlike, say, DP).
* Comment on the AMG method referenced by 8m3n, and their differences.
* Stress on how the quality of your produced samples and compare to those in competing methods.
* Stress on the limitations of the assumption of the model size scaling linearly with the number of samples to be protected, as mentioned by Rev. t8KP. I agree with the authors that, in the scale studied here (up to 160K), this seems to be feasible, but it is unclear if it will remain feasible in one or two orders of magnitude higher.

**Additional Comments On Reviewer Discussion:**

This paper received comments from 4 reviewers. I'm summarizing their main points, and responses, here:

**6foL** (Score: 5, confidence: 4) Main concerns pertained to (i) the application of the applicability of the authors' theoretical insights and (ii) details of the experimental settings for the membership inference attacks and similarities to other methods. Both of these were addressed by the authors.

**8m3n** (Score: 3, confidence: 4) This reviewer is the most negative, recommending rejection. The reviewer details the following main concerns, on which I comment:
* Methodological simplicity is a weakness: the authors (and I) disagree.
* Access to training data or data to protect: access to this data seems unavoidable - please see my comments in the summary above.
* Not applicable to SOTA models: the authors (and I) disagree.
* Similarities to AMG: there are no further explanations provided by the reviewer (see recommendations above).
* Concerned about the quality of generated samples: authors push back, and I agree with them.

**t8KP** (Score: 5, confidence: 4) Besides recognizing the importance of the problem and the effectiveness of the results, the main concerns are:
* Scalability: The author demonstrate that their approach is effective in training on 160K samples (from data to be protected).
* Assumption of accessibility to data: see my comments on my Summary and recommendations.
* Scaling of the classifier: the Rev is concerned on the scaling of the classifier w.r.t. the amount of data to protect. This is a valid point. See my recommendations above.
* More experiments: the reviewer asks for more qualitative results, but this is unespectific. Authors: please try to include more visual results the produced samples.

**dYdC** (Score: 5, Conf: 4). The Rev stresses the method is "solid", and effective. The main concerns are:
* Motivation: the reviewer doubts that the problem under study is of concern. This is countered by the authors, as well as by the comments from other reviewers.
* Computational costs incurred in training the classifier: the authors argue that this is much cheaper than re-trainign the diffusion model, and I agree. The Rev. also seems mistaken about the applicability to SOTA models and on the amount of data experimented with.

---

### Decision · Program_Chairs · 2025-01-22

Accept (Poster)